# A STAT3-based gene signature stratifies glioma patients for targeted therapy

Melanie Si Yan Tan [1,2,11], Edwin Sandanaraj [1,2,3,11], Yuk Kien Chong[1], See Wee Lim[1], Lynnette Wei Hsien Koh[1,2], Wai Hoe Ng[4,5], Nguan Soon Tan [2,6,7], Patrick Tan[5,8], Beng Ti Ang[3,4,5,9,12] & Carol Tang[1,5,10,12]

Intratumoral heterogeneity is a hallmark of glioblastoma (GBM) tumors, thought to negatively influence therapeutic outcome. Previous studies showed that mesenchymal tumors have a worse outcome than the proneural subtype. Here we focus on STAT3 as its activation precedes the proneural-mesenchymal transition. We first establish a *STAT3* gene signature that stratifies GBM patients into *STAT3*-high and -low cohorts. STAT3 inhibitor treatment selectively mitigates *STAT3*-high cell viability and tumorigenicity in orthotopic mouse xenograft models. We show the mechanism underlying resistance in *STAT3*-low cells by combining *STAT3* signature analysis with kinome screen data on STAT3 inhibitor-treated cells. This allows us to draw connections between kinases affected by STAT3 inhibitors, their associated transcription factors and target genes. We demonstrate that dual inhibition of IGF-1R and STAT3 sensitizes *STAT3*-low cells and improves survival in mice. Our study underscores the importance of serially profiling tumors so as to accurately target individuals who may demonstrate molecular subtype switching.

[1] Neuro-Oncology Research Laboratory, Department of Research, National Neuroscience Institute, Singapore, Singapore. [2] School of Biological Sciences, Nanyang Technological University, Singapore, Singapore. [3] Singapore Institute for Clinical Sciences, Agency for Science, Technology and Research (A*STAR), Singapore, Singapore. [4] Department of Neurosurgery, National Neuroscience Institute, Singapore, Singapore. [5] Duke-National University of Singapore Medical School, Singapore, Singapore. [6] Institute of Molecular and Cell Biology, Agency for Science, Technology and Research (A*STAR), Singapore, Singapore. [7] Lee Kong Chian School of Medicine, Nanyang Technology University, Singapore, Singapore. [8] Cancer Science Institute of Singapore, National University of Singapore, Singapore, Singapore. [9] Department of Physiology, Yong Loo Lin School of Medicine, National University of Singapore, Singapore, Singapore. [10] Division of Cellular and Molecular Research, Humphrey Oei Institute of Cancer Research, National Cancer Centre, Singapore, Singapore. [11] These authors contributed equally: Melanie Si Yan Tan, Edwin Sandanaraj. [12] These authors jointly supervised this work: Beng Ti Ang, Carol Tang. Correspondence and requests for materials should be addressed to B.T.A. (email: ang.beng.ti@singhealth.com.sg) or to C.T. (email: carol_tang@nni.com.sg)

Patients with glioblastoma (GBM) frequently survive no >15 months despite surgical intervention with chemotherapy and radiation[1]. Tumor recurrence and the development of resistance toward standard-of-care treatment regimens are key reasons for the poor outcome and have been attributed to the cellular and molecular heterogeneity of tumor tissue[2–5]. Over the past decade, several large, publicly funded efforts demonstrated that gene expression drives brain tumor disease progression and clinical outcome[3]. Primary adult GBM tumors that are predominantly isocitrate dehydrogenase 1—wild type (IDH-WT) are subdivided into three glioma-intrinsic (GI) transcriptomic subtypes (proneural, classical, and mesenchymal) after separating out microglial- and stromal cell-type contribution[4]. The revised World Health Organization (WHO) classification scheme for brain tumors incorporates these molecular markers to influence treatment decision[1]. These efforts highlight the necessity to prescribe treatment regimens based on a stratified population.

We focused on signal transducers and activators of transcription (STAT3) where its activation has been demonstrated to effect a transition in molecular subtype to the aggressive mesenchymal profile[6]. The proneural–mesenchymal transition (PMT) process has been associated with recurrent tumors and, more recently, chemotherapeutic and radiation resistance, thought to arise from the selection of transitional glioma-initiating cell clones harboring gain in PMT transcriptomic patterns[6]. Thus targeting the STAT3 signaling axis is pivotal in disease management.

The interleukin 6/Janus kinase/STAT3 (IL-6/JAK/STAT3) pathway is involved in the pathogenesis of many human malignancies. Increased IL-6 levels are also found in conditions associated with inflammation, such as rheumatoid arthritis and inflammatory bowel disease and increasingly in hematological disorders and solid tumors, such as GBM[7]. In cancer, increased IL-6 levels result in hyperactivation of JAK/STAT3 signaling, typically associated with poorer prognosis[8–11]. In most myeloproliferative cancers, the genes encoding JAK enzymes, particularly JAK2, frequently contain gain-of-function mutations[12]. However, no such mutations can be detected in GBM tumors, thus implicating other mechanisms of STAT3 activation. STAT3 has also been shown to regulate the self-renewal potential of glioma cells, suggesting that its inhibition would lead to a more curative and sustained outcome[13,14]. Over the past decade, much effort has been spent at evaluating JAK inhibitors predominantly in chronic inflammation and hematological disorders, with its application in solid tumors largely unexplored. Tofacitinib, Ruxolitinib, and pacritinib are the most advanced drugs in preclinical development. Furthermore, oncology-based clinical trials of STAT3 inhibitors have yet to take into consideration the implications arising from The Cancer Genome Atlas (TCGA) findings and whether the development of subsequent patient stratification methods would lead to significant improvement in prognostic outcomes. We are therefore interested in assessing JAK/STAT3-inhibitory agents in GBM tumors, with an ultimate goal in identification of potential responders and non-responders.

TCGA efforts identified molecular subtypes driven by key signaling pathways[3]. In the evaluation of upstream kinases that lead to active STAT3 signaling, the human epidermal growth factor receptor (EGFR), a member of the ErbB/HER family of receptor tyrosine kinases (RTKs); the family of IL-6–type (IL-6) cytokine receptors that form complexes with gp130 and JAKs; and several G-protein-coupled receptors have been described in previous literature[15]. Multiple growth factors (e.g., EGF, transforming growth factor-α (TGFα), platelet-derived growth factor, and colony-stimulating factor 1) and cytokines (e.g., insulin-like growth factor 1 (IGF-1), IL-6, leukemia inhibitory factor, cardiotrophin-1, ciliary neurotrophic factor, IL-10, IL-11, and Oncostatin-M) have been shown to activate STAT3. Elevated levels of STAT3-activating ligands, such as IGF-1, TGFα, or IL-6, have also been detected in the serum and/or the tumor microenvironment of patients with various malignancies. In the case of IGF-1 receptor (IGF-1R) where several small-molecule candidates are under evaluation in pharmaceutical pipelines, the downstream cell-intrinsic activation of STAT3 remains unclear in GBM tumors.

We first combine gene candidates inversely implicated in the STAT3 response pathway in patient-derived GBM cells in the presence of STAT3 knockdown (KD), with candidates regulated in similar direction across the STAT3 axis in large, public clinical databases. This strategy allows us to prioritize clinically relevant gene candidates in an otherwise statistically underpowered cell line collection, as with all such studies. We then systematically rank the STAT3 signaling axis, as defined by a gene signature, with key patient characteristics and clinical indicators[16,17]. This allows us to predict patient cohorts most likely to benefit from a STAT3 inhibition therapeutic approach. Furthermore, by analyzing the upregulated genes in the other non-responder cohort, we select key kinases for which inhibitory small molecules are currently evaluated in clinical trials. To substantiate our bioinformatical analyses, we prioritize clinically relevant (and biochemically active) kinases using a novel computational pipeline to set the threshold for a kinome screen conducted on STAT3-perturbed GBM cells. We successfully identify IGF-1R in an as yet undescribed STAT3-IGF-1 forward feedback loop. Our study provides preclinical evidence for the implementation of anti-STAT3 therapy in selected patient cohorts, while defining a method to sensitize non-responders.

## Results

**STAT3 functionally tuned gene signature.** Brain tumor gene expression drives disease progression and patient survival outcome[4], suggesting that druggable pathways may be revealed through genomic and transcriptomic profiles. STAT3 represents the final molecular switch that is activated prior to the PMT process that typifies highly aggressive and recurrent GBMs[6]. We hypothesize that the STAT3 pathway stratifies patients for their likely response to STAT3 inhibition therapy. As any signaling pathway is better represented by a set of genes than a single candidate, we established a transcriptomic signature reflecting the STAT3 pathway activation status (Supplementary Data 1). We prioritized genes that contribute functionally to the STAT3 pathway and correlate with prognostic outcome. STAT3 co-expressed genes from the Rembrandt patient database (Fig. 1a, middle panel) that displayed inverse expression upon STAT3 KD in patient-derived GBM-propagating cells (GPCs; Fig. 1a, left panel) were identified to form the STAT3 "functionally tuned" gene signature (Fig. 1a, right panel)[18]. The latter approach ensures that only genes downstream and modulated by the STAT3 pathway would be selected. We verified STAT3 protein expression upon lentiviral-mediated KD in three GPCs and observed significant mitigation of viability, sphere-forming frequency, and sphere size (Supplementary Fig. 1a–l). We established a positive enrichment of the JAK/STAT signaling pathway in our functionally tuned gene signature, and defined it as STAT3-high, while an inverse correlation defined the STAT3-low gene signature (Supplementary Fig. 1m; Supplementary Data 2). We then tested the robustness of our STAT3 composite signature in two clinical databases, Gravendeel and TCGA (Gravendeel, Fig. 1b–f; TCGA, Supplementary Fig. 2a, b)[3,19]. Accordingly, contingency analyses accounting for TCGA GI molecular subtypes and the WHO classification scheme including molecular and clinical indicators, demonstrate that STAT3-high defines a patient cohort enriched in the mesenchymal and classical

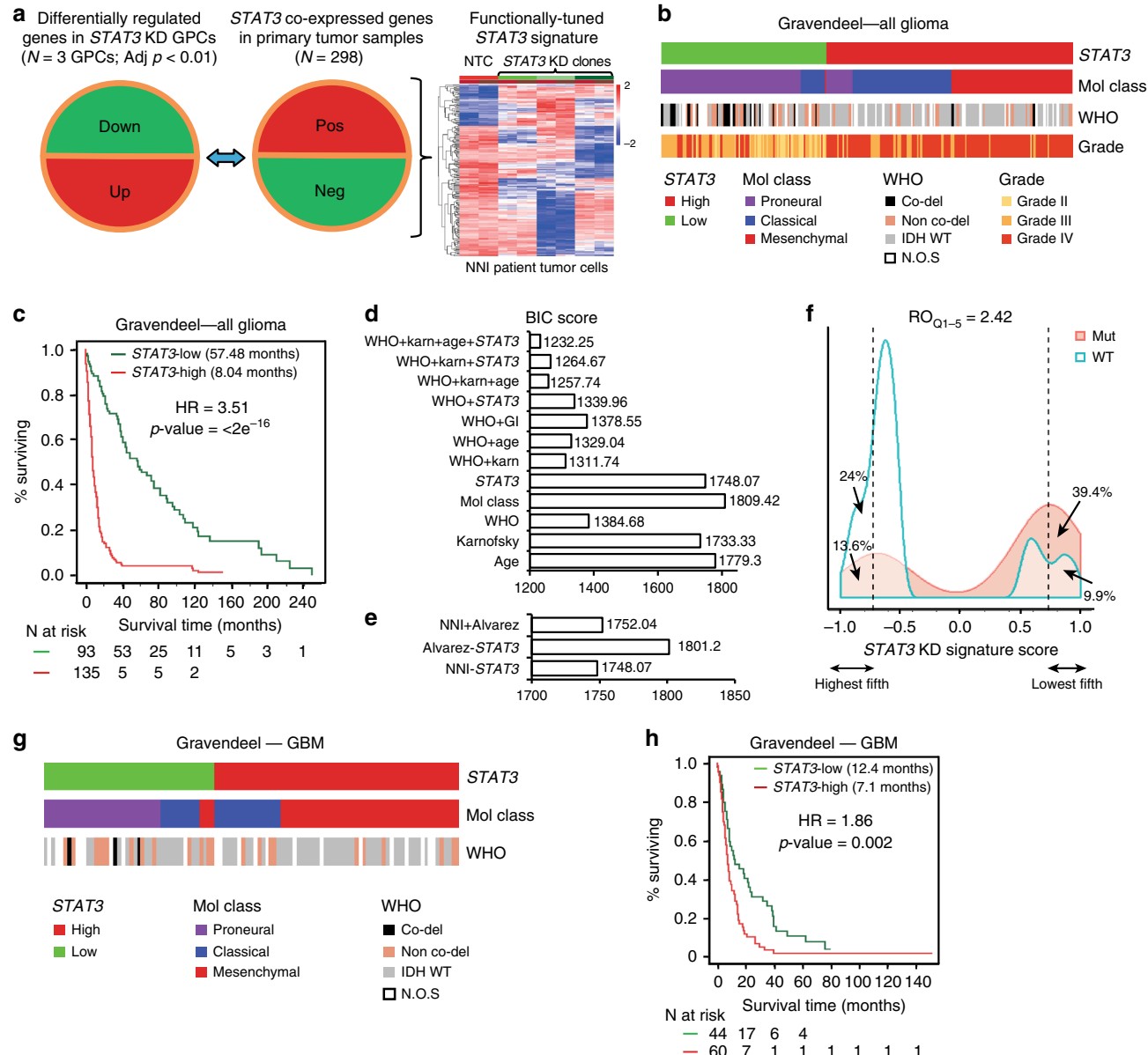

**Fig. 1** NNI-*STAT3* functionally tuned gene signature stratifies patient survival independent of current clinical indicators. **a** *STAT3* co-expressed genes from Rembrandt patient database (middle panel) that displayed inverse expression upon *STAT3* knockdown (KD) (left panel) were identified to form the NNI-*STAT3* functionally tuned gene signature (right panel). **b** In Gravendeel clinical database, *STAT3*-high patient all glioma cohort was enriched in mesenchymal and classical molecular subtypes, with predominantly isocitrate dehydrogenase 1 (IDH)-wild-type status. *STAT3*-low tumors, in contrast, comprised mostly low-grade gliomas (LGGs), IDH-mutant (1p/19q co-deleted and non-co-deleted), and proneural molecular subtypes. **c** NNI-*STAT3* signature stratified all glioma patient survival in Gravendeel clinical database. An enrichment of *STAT3* pathway activation defined the poor prognosis patients (*STAT3*-high, 8.04 months), while patients of *STAT3*-low survived significantly longer (57.48 months). **d** A combination of NNI-*STAT3* gene signature, World Health Organization status, Karnofsky (Karn) score, and age presented the best statistical model to account for the variability in patient survival, using the Bayesian Information Criterion (BIC) method. **e** NNI-*STAT3* signature performed better than the existing Alvarez *STAT3* signature for glioma patient prognosis. **f** The relative odds of correlation between *STAT3* signature and IDH mutation is 2.42 in a diagnostic metrics test. Patients with a negative signature score (*STAT3*-high) are 2.42 times more likely to be IDH-wild-type than those with a positive signature score (*STAT3*-low). **g** *STAT3*-high GBM patient cohort was enriched in mesenchymal and classical molecular subtypes. *STAT3*-low tumors, in contrast, comprised mostly the proneural molecular subtype. **h** NNI-*STAT3* signature stratified glioblastoma patient survival. An enrichment of *STAT3* pathway activation defined the poor prognosis patients (*STAT3*-high, 7.1 months) while patients of *STAT3*-low survived significantly longer (12.4 months)

molecular subtypes, typifying highly aggressive and recurrent gliomas (Supplementary Table 1a). These tumors also demonstrated a significant enrichment of 1p/19q non-co-deletion and IDH-WT status. *STAT3*-low tumors, in contrast, comprise mostly of low-grade gliomas (LGGs), and the proneural molecular subtype with enrichment of 1p/19q co-deletion and IDH-Mut (mutant) status, representing tumors of better prognosis and

responsiveness to current chemotherapy (Fig. 1b, and additional clinical database TCGA, Supplementary Fig. 2a)[20]. Figure 1c demonstrates patient survival stratification based on our "functionally tuned" *STAT3* gene signature. *STAT3*-high defines poor prognosis patients, while *STAT3*-low patients survived significantly longer (logrank *p* value < $2 \times 10^{-16}$) (additional clinical database, TCGA; Supplementary Fig. 2b). Further univariate and

multivariate analyses suggested that the *STAT3* signature functions as an independent predictor and is not confounded by current molecular and clinical indicators (Supplementary Table 1b). We demonstrate using the Bayesian Information Criterion (BIC) method that a combination of *STAT3*, the WHO classification system that incorporates the IDH and 1p/19q co-deletion status, and Karnofsky score (measures patient's functional status) presented the best statistical model accounting for patient survival (Fig. 1d). In such plots, a reduction in the BIC score by an absolute value of 10 fulfills the industry standard for advancing a therapeutic strategy into clinical trial[21]. In addition, our *STAT3* signature outperformed the existing Alvarez *STAT3* gene signature previously established to be a pan-solid, tumor-specific profile for glioma patient prognosis (Fig. 1e)[22]. The relative odds of correlation between *STAT3* signature and IDH mutation is 2.42 in a diagnostic metrics test. Patients with a negative signature score (*STAT3*-high) are 2.42 times more likely to be IDH-WT than those with a positive signature score (*STAT3*-low, IDH-Mut) (Fig. 1f).

As GBM patients portend the poorest prognosis over a decade, with little improvement even with the best standard-of-care drug temozolomide (TMZ), we extended our analyses to exclusively GBM tumors in both Gravendeel and TCGA databases (Fig. 1g, h and Supplementary Fig. 2c, d). Similar prognostic association was observed in GBM patients for *STAT3*-high and -low subtypes (Gravendeel, logrank *p* value = 0.002, Fig. 1h; TCGA, logrank *p* value = 0.009; Supplementary Fig. 2d). *STAT3*-high significantly enriched for the mesenchymal and classical subtypes, with predominantly IDH-WT and 1p/19q non-co-deletion status (Gravendeel, Fig. 1g; TCGA, Supplementary Fig. 2c; Supplementary Table 1b). Taken together, our data suggest that the *STAT3* pathway contributes to the molecular heterogeneity of GBM tumors.

**STAT3-high group shows improved response to STAT3 inhibitors**. We stratified our GPCs based on our *STAT3* gene signature and observed a consistent expression of increased phospho-STAT3 (pSTAT3) in the *STAT3*-high group (NNI-21, -24, -12), in contrast to the *STAT3*-low cells (NNI-11, -20 and -23) (Fig. 2a and Supplementary Fig. 3). *STAT3*-high tumor cells demonstrated significantly lower half maximal inhibitory concentrations (IC$_{50}$) upon treatment with STAT3 inhibitors, compared to *STAT3*-low cells (Fig. 2b–d and Supplementary Fig. 3c–e). AZD1480, Stattic, and WP1066 represent JAK/STAT inhibitors commonly used; in particular, AZD1480 has been shown to exhibit specific activity against Jak2 kinase, mitigating tumor cell proliferation in a variety of solid tumors[23]. *STAT3*-high cells also showed reduced cell viability and gliomasphere-forming ability when compared to *STAT3*-low cell lines (Fig. 2e–j). Furthermore, using a recovery assay that allows treated cells to recover in the absence of the drug to ascertain prolonged and irreversible inhibition, *STAT3*-high cells showed significant mitigation of viability, self-renewal, and invasive potential (Fig. 2k–m). In contrast, *STAT3*-low cells were minimally inhibited by the STAT3 inhibitors and instead developed resistance. Finally, utilizing the orthotopic patient-derived xenograft (PDX) mouse model, mice implanted with AZD1480-treated NNI-24 (*STAT3*-high) cells demonstrated better survival in a dose-dependent manner (Fig. 2n and Supplementary Table 2, Kaplan–Meier statistics). Compared to NNI-20 (*STAT3*-low) mice, NNI-24 mice demonstrated an approximate 2.5-fold increased median survival difference between matched dimethyl sulfoxide (DMSO) solvent control and AZD1480-treated groups (Fig. 2o and Supplementary Table 2). All GPCs and PDX tumors were sequenced and defined as IDH-WT (Supplementary Fig. 3f,

g). These findings support the application of our *STAT3* gene signature to stratify and identify patient cohorts most likely to receive treatment benefit from STAT3 inhibition therapy, while further cautioning against the use of such inhibitors in *STAT3*-low patients due to the development of resistance mechanisms.

**IGFBP2 causes chemoresistance in STAT3-low glioma cells**. As important as identifying potential responders, we explored mechanisms underlying resistance in the *STAT3*-low group, so that therapeutic options may be defined to sensitize these individuals to chemotherapy. We first evaluated our *STAT3* functionally tuned gene signature and prioritized those candidates most highly variable between *STAT3*-high and *STAT3*-low groups. Since upregulated genes better serve as therapeutic targets, we focused on candidates that exhibited a dose-dependent increase in *STAT3*-low cells after treatment with AZD1480 (Fig. 3a). These same candidates were then verified to display an inverse pattern (i.e., dose-dependent reduction) in the *STAT3*-high cells subjected to similar treatment, for the reason that this candidate list should exhibit differential expression between the two stratified GPC groups (Supplementary Data 3). Similar results were obtained with Stattic and WP1066 treatment, thus supporting the specificity of targeting the STAT3 signaling axis (Supplementary Fig. 4a, b). Six genes were identified, namely, insulin-like growth factor binding protein 2 (*IGFBP2*), neural precursor cell expressed, developmentally downregulated 9 (*NEDD9*), synaptosomal-associated protein 23 (*SNAP23*), guanosine diphosphate (GDP)-mannose pyrophosphorylase A (*GMPPA*), E26 transformation-specific containing gene (*ELK3*, ETS domain containing protein), and kelch domain containing 8A (*KLHDC8A*). By surveying literature, we prioritized *IGFBP2* as it is one of the six similar genes that sequester intracellular IGF-1 and for which clinical trials are currently in progress to evaluate anti-IGF-1R inhibitors in a variety of solid tumors. Since kinases represent dominant therapeutic targets in major pharmaceutical pipelines, we established the approach of using biological evidence to substantiate our computational predictions, by measuring phosphorylation levels of 144 kinases in *STAT3*-signature-stratified GPCs using the PamChip kinome screen. We developed a novel computational pipeline on kinome assay data by integrating phospho-chemical interactions with functional genomics data through kinase-substrate databases (Supplementary Fig. 4c). Briefly, the phosphorylation dynamics of kinase substrates were measured as quantitative readouts. These peptide readouts were mapped with relevant kinases using protein databases such as Kinexus, PhosphoSite, Reactome, and human protein reference databases[24–26]. We calculated the quantitative summary of kinase activity using a rank-based clustering method as described in the "Methods" section. Subsequently, the differential kinase regulation upon AZD1480 treatment was estimated using the linear regression model[27]. Our approach confirms IGF-1R as a top-ranking tyrosine kinase uniquely and biochemically elevated in *STAT3*-low tumors upon treatment with AZD1480 (Fig. 3b and Supplementary Table 3). Treatment of GPCs with AZD1480 demonstrated an increase in pSTAT3 and IGFBP2 expression in the nuclear fraction of *STAT3*-low but not of *STAT3*-high cells, consistent with *IGFBP2* as a target of STAT3 transcription factor (Fig. 3c). We verified a dose-dependent increase of secreted IGFBP2 and IGF-1 protein in NNI-20 cells treated with AZD1480 (*STAT3*-low GPC) (Fig. 3d). Furthermore, we observed a significant increase of IGFBP2 and IGF-1R proteins in *STAT3*-low cells (NNI-20, -23) upon AZD1480 treatment, compared to *STAT3*-high cells (NNI-21, -24) (Fig. 3e). Our data here provide a basis to exploring the IGF-1R pathway in *STAT3*-low cells.

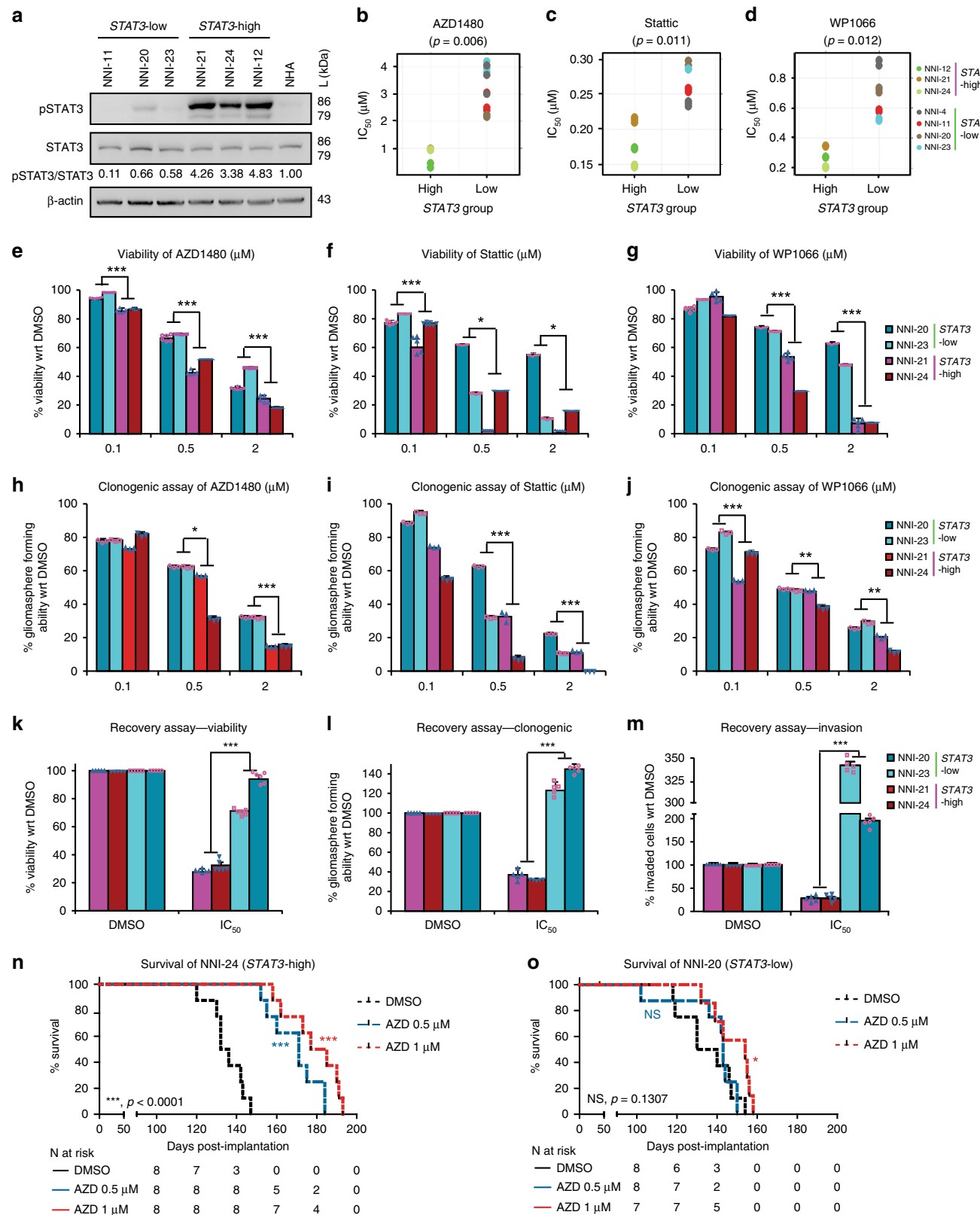

**Inhibition of STAT3 and IGF-1R sensitizes _STAT3_-low cells**.
We first attenuated IGF-1R signaling pathway by targeting its effector, _IGFBP2_, using lentiviral-mediated short hairpin RNA (shRNA) KD (NNI-20, Fig. 4a–c; NNI-23, Supplementary Fig. 5a–c). KD of _IGFBP2_, resulting in IGFBP2 protein reduction, significantly sensitized _STAT3_-low cells to STAT3 inhibitor treatment, demonstrated by decreased viability and clonogenic potential. More importantly, _IGFBP2_ KD in _STAT3_-low cells sensitized them to AZD1480 and mitigated their viability and self-renewal capacity. This suggests that dual targeting of STAT3 and the IGF-1R/IGFBP2 signaling axis presents a viable therapeutic strategy to abolish resistance in _STAT3_-low GPCs.

**Fig. 2** NNI patient-derived glioblastoma (GBM) cells stratified by their *STAT3* status show variable response to signal transducers and activators of transcription 3 (STAT3) inhibitors. **a** Immunoblot analysis of patient GPCs. *STAT3*-high cell lines showed elevated phospho-STAT3, compared to *STAT3*-low cell lines. **b–d** Patient GBM cells were treated with **b** AZD1480, **c** Stattic, and **d** WP1066, and their IC$_{50}$ values were determined (results are mean of triplicate experiments). Consistent with bioinformatical prediction, *STAT3*-high cell lines showed sensitivity to STAT3 inhibitors as demonstrated by lower IC$_{50}$ values. *STAT3*-high and -low cell lines were validated by **e–g** cell viability and **h–j** clonogenic capacity after treatment with STAT3 inhibitors. *STAT3*-low cells demonstrated greater viability and gliomasphere-forming capability after treatment with **e, h** AZD1480, **f, i** Stattic, and **g, j** WP1066. Conversely, *STAT3*-high lines displayed greater sensitivity to STAT3 inhibitors, resulting in reduced cell viability and gliomasphere-forming capability. **k–m** Recovery assay after 5-day AZD1480 treatment, and the **k** viability and **l** clonogenic capacity of *STAT3*-high GPCs were significantly mitigated. In contrast, *STAT3*-low GPCs developed resistance and **m** demonstrated greater ability to invade. *$p < 0.05$; **$p < 0.01$; ***$p < 0.001$; *STAT3*-high versus *STAT3*-low. For statistical analysis, two-sided Student's *t* test was used. Error bars represent standard deviation of the mean. All results are mean of triplicate experiments. **n** Orthotopic tumors established from *STAT3*-high, AZD1480-pretreated cells resulted in mice with prolonged survival. *STAT3*-high (47 days) patient-derived xenograft demonstrated greater median survival difference of ~2.5-fold for AZD1480 arm compared to **o** NNI-20 (19 days) animal groups. *$p < 0.05$; ***$p < 0.001$ versus dimethyl sulfoxide. For statistical analysis, logrank test was used

To effectively target the STAT3/IGFBP2/IGF-1/IGF-1R feed-forward mechanism in the *STAT3*-low GPCs, we conducted dual inhibition of both STAT3 and IGF-1R pathways using AZD1480 and Linsitinib, an experimental drug candidate targeting IGF-1R used in various malignancies. This strategy demonstrated synergistic effect in two *STAT3*-low cells in vitro and in vivo (NNI-20, Fig. 4d–h; NNI-23, Supplementary Fig. 5d–f). We observed a significant dose-dependent reduction of cell viability and self-renewal upon single agent treatment alone (approximately 55% viability) but greatly improved synergistic outcome in the presence of both AZD1480 and Linsitinib (up to approximately 10% viability). We subsequently generated orthotopic xenografts to evaluate tumorigenicity using these same pretreated GPC lines where, as expected, dual inhibition of STAT3 and IGF-1R conferred the greatest survival benefit and extended tumor latency in the *STAT3*-low group of mice (Fig. 4g, h). Protein expression of pSTAT3, total STAT3, IGFBP2, and IGF-1R were verified in these xenografted tissues (Fig. 4i, j and Supplementary Fig. 5g–l). We propose that, in *STAT3*-low cells, phosphorylated STAT3 activates *IGFBP2* transcription, which increases the production of IGF-1. This in turn triggers the activation of the IGF-1R pathway, contributing to an as yet undescribed feed-forward mechanism (Supplementary Fig. 6). These results suggest a potential therapeutic strategy utilizing a dual inhibitor approach to sensitize the *STAT3*-low GBM patient subgroup.

**AZD1480 and/or Linsitinib synergize with TMZ.** To assess the efficacy of our proposed therapeutic strategies in either *STAT3*-high or -low patient cohorts, we first treated both *STAT3*-stratified GBM cells with DMSO solvent, 0.5 μM AZD1480 with/without TMZ at 20–200 μM range (Fig. 5a). These in vitro TMZ concentrations are routinely used in literature[28–30]. We observed significant dose-dependent mitigation of GBM cell viability in the presence of AZD1480 and 50–200 μM TMZ. In contrast, *STAT3*-low GBM cells demonstrated a marginal, <20% decrease in viability (albeit significant) with AZD1480 and 20–200 μM TMZ. We evaluated the combination index (CI) plot where increased synergism with TMZ correlated with CI values of <1 (Fig. 5b). The CI values were calculated using the CompuSyn software for evaluation of drug combinations[31,32].

Next, we assessed the fraction affected (i.e., reduced viability) in *STAT3*-low cells after treatment with AZD1480, Linsitinib, or both (Fig. 5c). Similarly, our CI plot showed increasing synergism with TMZ, as denoted by the decreasing CI value (Fig. 5e). We also carried out similar AZD1480 and TMZ treatment in *STAT3*-low cells with *IGFBP2* KD for definitive mechanistic implication (Fig. 5d). The rationale arose from our earlier data showing that STAT3 activation leads to *IGFBP2* gene transcription, which in turn stimulates the production of IGF cytokine (Supplementary

Fig. 6). Our CI plot demonstrated synergism with TMZ at 50–200 μM range (Fig. 5f).

Collectively, our in vitro data provide strong evidence for both STAT3 inhibition and dual STAT3/IGF-1R inhibition in *STAT3*-high and -low GBM cells, respectively, and synergize with TMZ, thus suggesting the advancement of both therapeutic approaches in a clinical setting.

To provide further support of our in vitro data, we focused on demonstrating that STAT3 inhibitors can selectively target *STAT3*-high GBM tumors. The premise of our approach lies in TCGA studies showing that transcriptomic expression drives GBM disease progression and prognostic outcome[3,4]. We tapped into a recent article where drug and disease signature integration identifies synergistic combinations in GBM[33]. This study utilized the Library of Integrated Network-Based Cellular Signatures (LINCS) database where several commercial cancer cell lines were treated with Food and Drug Administration-approved and experimental small molecule drugs, and the transcriptomic profile of each treated cell line was acquired. Primary GBM cells were similarly treated with these drugs including TMZ with radiation for the purpose that synergistic interactions could be assessed with the standard-of-care treatment regimen for GBM patients. The authors further mapped the association of transcriptomic patterns to prognostic information in TCGA, thus identifying clinically relevant drug combinations capable of reversing the disease transcriptomic profile. In our specific scenario, the disease pattern is defined by our *STAT3* functionally tuned gene signature for which we previously demonstrated phenotypic effects and prognostic association (Figs. 1 and 2 and Supplementary Fig. 3a, b). Thus, using an orthogonal plot, we identified drugs that demonstrated low concordance with TMZ and high discordance with the *STAT3*-high tumor phenotype (Fig. 5g). This indicates that the drug acts in a synergistic manner with TMZ and is capable of reversing the disease transcriptomic profile. Interestingly, Ruxolitinib, a Jak2 inhibitor, and AZD1480 emerged in the top ranked drugs (Supplementary Table 4). These drugs thus have the potential to reverse the *STAT3*-high disease profile and support their use in targeting the PMT process that typifies highly aggressive and recurrent tumors.

**Profiling recurrent tumors prior to treatment administration.** Conventional methods to detect STAT3 pathway activation are by immunohistochemistry where pSTAT3-specific antibodies are used on frozen or paraffin-embedded tumor sections. We provide data that our *STAT3* functionally tuned gene signature outperforms pSTAT3 status alone. We showed that pSTAT3 staining in our NNI patient tumors was inadequate to stratify the *STAT3*-high and -low patient groups, for which we previously demonstrated significant correlation with IC$_{50}$ values (Fig. 2b–d and Fig. 6a). Briefly, comparing various methods of *H*-score,

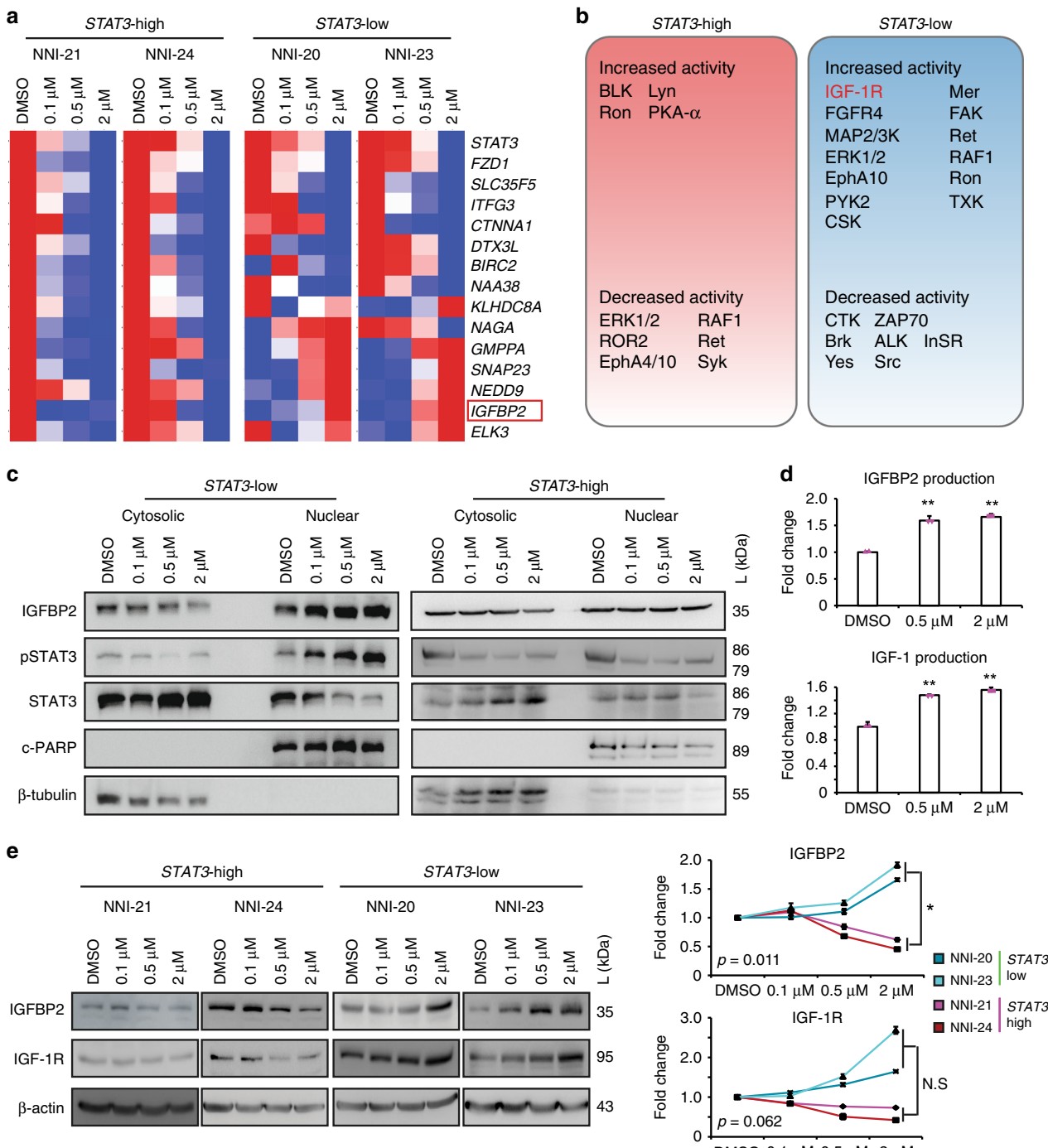

**Fig. 3** Mechanistic gene candidates identified by NNI-*STAT3* gene signature. **a** Winnowed gene list across patient tumors identified candidates uniquely upregulated in *STAT3*-high tumors. A dose-dependent differential gene expression after signal transducers and activators of transcription 3 (STAT3) inhibitor AZD1480 treatment distinguished cooperative genes responsible in the *STAT3*-resistant profile. Results are mean of triplicate experiments. **b** Graphical illustration of responsive and resistant protein tyrosine kinase candidates in *STAT3*-high and -low cell lines treated with AZD1480 using computational workflow described in Supplementary Fig. 4c. **c** Treatment of cells with STAT3 inhibitor (AZD1480) demonstrated an increase in pSTAT3 and insulin-like growth factor binding protein 2 (IGFBP2) expression levels in the nuclear fraction of *STAT3*-low cells but not in *STAT3*-high. **d** Fold change differences in secreted proteins demonstrated increased IGFBP2 and insulin-like growth factor 1 receptor (IGF-1R) in resistant cells post-treatment with STAT3 inhibitor. Results are mean of triplicate experiments. **e** *STAT3*-high glioblastoma (GBM) cells displayed modest reduction in IGF-1R and IGFBP2 expression levels. In contrast, IGF-1R and IGFBP2 protein expression in *STAT3*-low cells increased dose-dependently upon AZD1480 treatment, albeit IGF-1R was marginally insignificant. Fold change differences in protein expression of IGFBP2 and IGF-1R were compared between *STAT3*-high and -low GBM cells. *$p < 0.05$; **$p < 0.01$; *STAT3*-high versus *STAT3*-low. For statistical analysis, two-sided Student's $t$ test was used. Error bars represent standard deviation of the mean

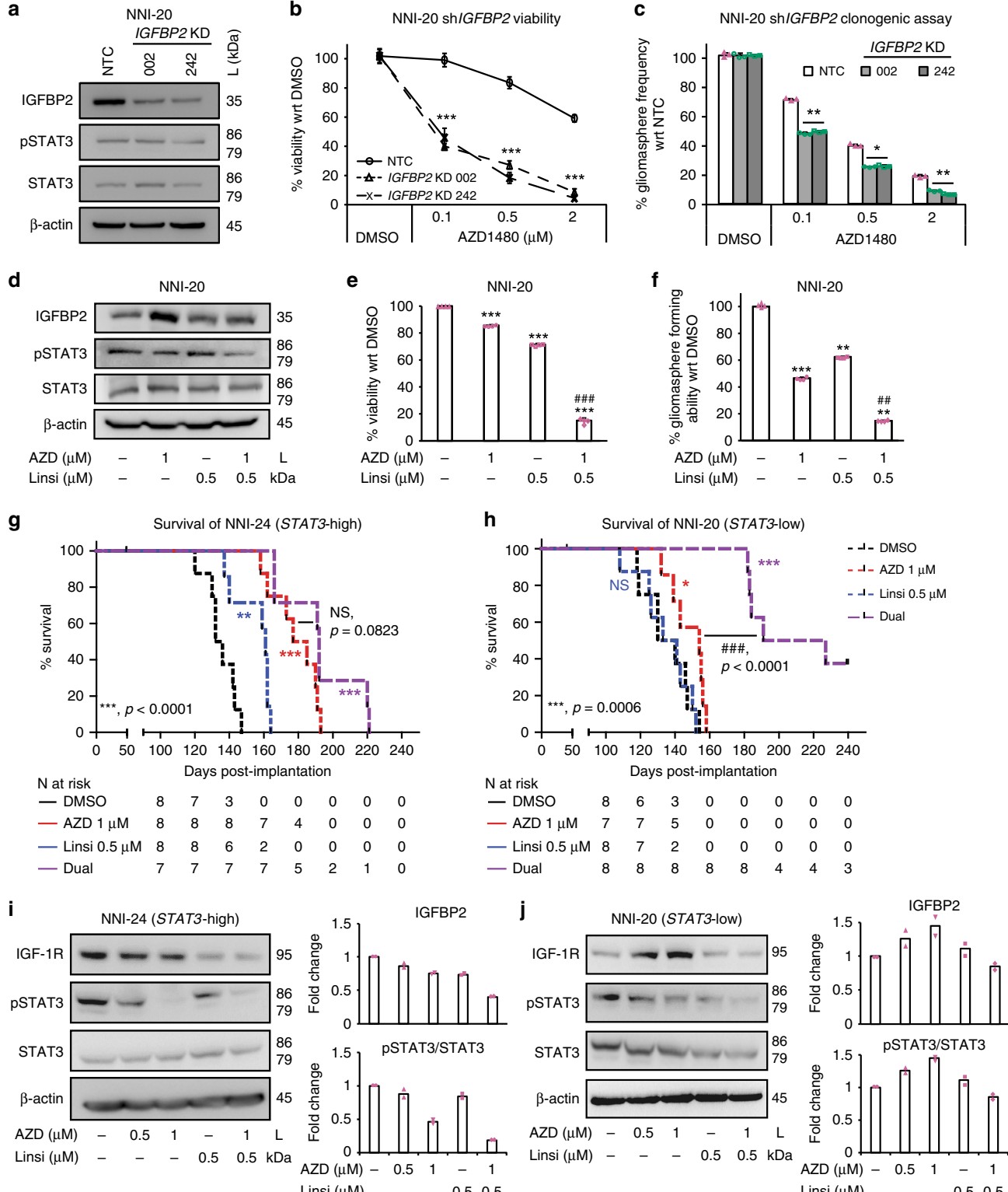

IC$_{50}$ values, and our gene signature score (Nearest Template Prediction (NTP) score), we showed that there was no significant correlation of *H*-score to either the NTP score or IC$_{50}$ values (Fig. 6b, c and Supplementary Fig. 7). However, when we compared the IC$_{50}$ values to the NTP score, we derived a significant negative correlation (Fig. 6d), suggesting that the *STAT3* functionally tuned gene signature is able to accurately profile sensitive and resistant patient cohorts.

To assess the utility of STAT3 inhibitors in recurrent patient tumors, a likely clinical scenario where individuals may be prescribed novel agents, and when PMT characterizes aggressive recurrence, we evaluated TCGA glioma-intrinsic subtypes (GISs) with STAT3 status as defined by the *STAT3* gene signature[4]. Our previous data showed that *STAT3*-high enriched for almost equal patient numbers of mesenchymal and classical subtypes in either of glioma or GBM cohort (Gravendeel, Fig. 1b, g; TCGA,

**Fig. 4** Sensitization of chemoresistant *STAT3*-low glioma cells by *IGFBP2* knockdown (KD) and dual drug inhibition. **a** Depletion of mechanistic gene *IGFBP2*. Compared to the non-targeting control (NTC), sh*IGFBP2* clones displayed increased sensitivity to AZD1480, observed by **b** decreased viability and **c** reduced gliomasphere-forming frequency in *STAT3*-low cell line, NNI-20 (additional cell line NNI-23 provided in Supplementary Fig. 5a–c). **p** < 0.01; ***p* < 0.001; KD versus NTC. For statistical analysis, two-sided Student's *t* test was used. Error bars represent standard deviation of the mean. **d–f** Using a dual drug treatment strategy (AZD1480 against signal transducers and activators of transcription 3 (STAT3), and Linsitinib against insulin-like growth factor 1 receptor (IGF-1R)), NNI-20 demonstrated a reduction of IGF-1R and pSTAT3 as observed in **d** immunoblot analysis, **e** viability, and **f** gliomasphere-forming frequency assays. Additional cell line, NNI-23, is provided in Supplementary Fig. 5d–f. **p** < 0.01; ***p* < 0.001; treatment groups versus dimethyl sulfoxide (DMSO) control. ##*p* < 0.01; ###*p* < 0.001; dual inhibitors versus individual inhibitor (AZD1480 or Linsitinib). The combination index value for the combined drugs 1 μM AZD1480 and 0.5 μM Linsitinib is 0.2092, calculated using CompuSyn. **g** NNI-24 *STAT3*-high xenograft model established from AZD1480-pretreated glioblastoma (GBM) cells displayed prolonged survival, while **h** NNI-20 *STAT3*-low xenograft model received marginal, albeit significant (for AZD1480 only) survival benefit with single agent alone. In contrast, dual treatment targeting both STAT3 and IGF-1R significantly prolonged survival and extended tumor latency of *STAT3*-low patient-derived xenograft (PDX) mice. *p* < 0.05; **p** < 0.01; ***p* < 0.001; treatment group versus DMSO; ##*p* < 0.01; ###*p* < 0.001 dual inhibitors versus single inhibitor (logrank test). Censored points are indicated by the black tick mark, where mice death was not attributed to tumor formation. Immunoblot analysis of PDX tumors demonstrated that mice implanted with **i** NNI-24 (*STAT3*-high) treated with AZD1480, demonstrated a reduction in pSTAT3 expression, while **j** NNI-20 (*STAT3*-low) showed a stark increase in IGF-1R expression. This supports that dual inhibition of STAT3 and IGF-1R serves as a possible therapeutic strategy for *STAT3*-low GBM patients. Bar chart indicates quantified average fold change from immunoblots of two mice per treatment. This was limited by retrieval of sizeable tumors from dual treatment animal arm. Duplicate data are shown in Supplementary Fig. 5k–l

Supplementary Fig. 2a, c). We demonstrate that the mesenchymal patient cohort predominantly maintains its *STAT3*-high profile (86%), whereas the *STAT3*-low group underwent subtype switching to *STAT3*-high (100%) (Supplementary Fig. 8a). However, this pattern was not clearly marked in the non-mesenchymal cohort (classical, typifies gain-of-function *EGFR* mutations, and proneural). Specifically, classical *STAT3*-high patients had an equal chance to undergo subtype switching to *STAT3*-low. Although these results strongly support that mesenchymal patient groups can potentially benefit from STAT3 inhibition therapy, they also indicate that the dual inhibition strategy would most likely benefit patients who had undergone STAT3 inhibition therapy at first diagnosis but gained the resistant *STAT3*-low profile upon recurrence.

A molecularly well-annotated brain tumor resource to enable preclinical studies is of paramount importance. Our collection comprises GBM tumor tissue that compares with tumors acquired by TCGA (Supplementary Fig. 8b–f). We show that our xenograft tumors recapitulate the molecular subtypes of their patients' original tumors and are clustered with TCGA's proneural, classical, and mesenchymal groups. Furthermore, these tumors demonstrate similar enrichment of neural cell lineages previously associated with tumor-initiating and propagating capacity[2,5,34]. The tumor purity score of our resource is 62%, comparable with 59% of the TCGA collection (Supplementary Fig. 8g–i).

## Discussion

The utility of STAT3 inhibitors has largely been confined to myeloproliferative disorders, in part due to their poor blood–brain barrier (BBB) penetration. In polycythemia vera, the STAT3 pathway correlates with poorer prognosis and is constitutively active due to the presence of the *JAK2* V617F mutation[12]. However, no such mutation exists in GBM tumors, although STAT3 has been implicated in the proliferation and self-renewal of GBM stem-like cells[13,14]. We have also observed that increased *STAT3*-wild-type expression correlates with poor prognostic outcome. This suggests that other mechanisms of STAT3 pathway activation remains, and current STAT3 inhibitor molecules with efficient BBB penetration capability may find utility in GBM treatment. To add to the complexity of solid tumors, several studies have suggested the presence of molecular heterogeneity[3]. This may account for the frequently observed inter-patient variability to treatment response. Indeed, the mesenchymal profile has been associated with the poorest

prognosis, while the proneural subtype typifies the more sensitive and treatable cohort[4]. Currently, routine pathological diagnosis uses morphological features to define the grades of tumor tissue. We now know that such histological approaches are woefully inadequate to influence treatment decisions. Precision oncology applies these concepts of molecular markers and stratification to determine targeted therapeutic strategies.

We hypothesize that most signaling pathways, such as the IL-6/STAT3 axis, could be represented by a set of genes defining key regulatory modules. The premise of our hypothesis rests in being able to map these modules in clinical databases comprising molecular information and indicators used by the physician[35,36]. Such a strategy facilitates the quantitative analysis of multi-dimensional data represented as molecular information, magnetic resonance imaging scans, and clinical indicators used to assess the patient's disease and functional status. We previously demonstrated the utility of such a strategy in determining tumor cell resistance and invasiveness[37–41]. In this study, we identified *STAT3*-high to describe a cohort of both glioma and GBM patients who had poorer prognosis. This subgroup comprised of genes previously implicated in ATP-binding cassette (ABC) drug transporters, RTK signaling, and tumor cell invasiveness. Our method to winnow down the gene list associated with *STAT3* combined functional validation with co-expressed genes in clinical databases. This reduced our scope to only clinically relevant genes with phenotypic changes in *STAT3*-perturbed primary GBM cells and PDX mouse models. Our *STAT3* signature is not confounded by current clinical and molecular classification, thereby emphasizing the molecular heterogeneity contributed by this mechanistic pathway. While we showed significant extended survival after implanting AZD1480-pretreated *STAT3*-high GBM cells in immunocompromised mice, we also identified the top ranking causative mechanism responsible for conferring increased resistance after STAT3 inhibition therapy in *STAT3*-low patients and validated its biochemical activity using a kinome screen. GBM tumor cell resistance to targeted therapy is often attributed to the compensatory activation of RTKs[42–44]. Studies have described the frequent activation of insulin receptor (InsR) and IGF-1R in GBM specimens and PDX cells at conferring resistance to EGFR inhibitors[43–45], both frequently activated but rarely amplified or mutated in GBM according to TCGA (<2%)[46]. IGFBP2 is the second most abundant IGF-binding protein (after IGFBP3), functions as a carrier for IGF-1 and likely promotes tumor progression through IGF-1R pathway[47]. In gliomas, IGFBP2 is also often overexpressed[48]; moreover, increased

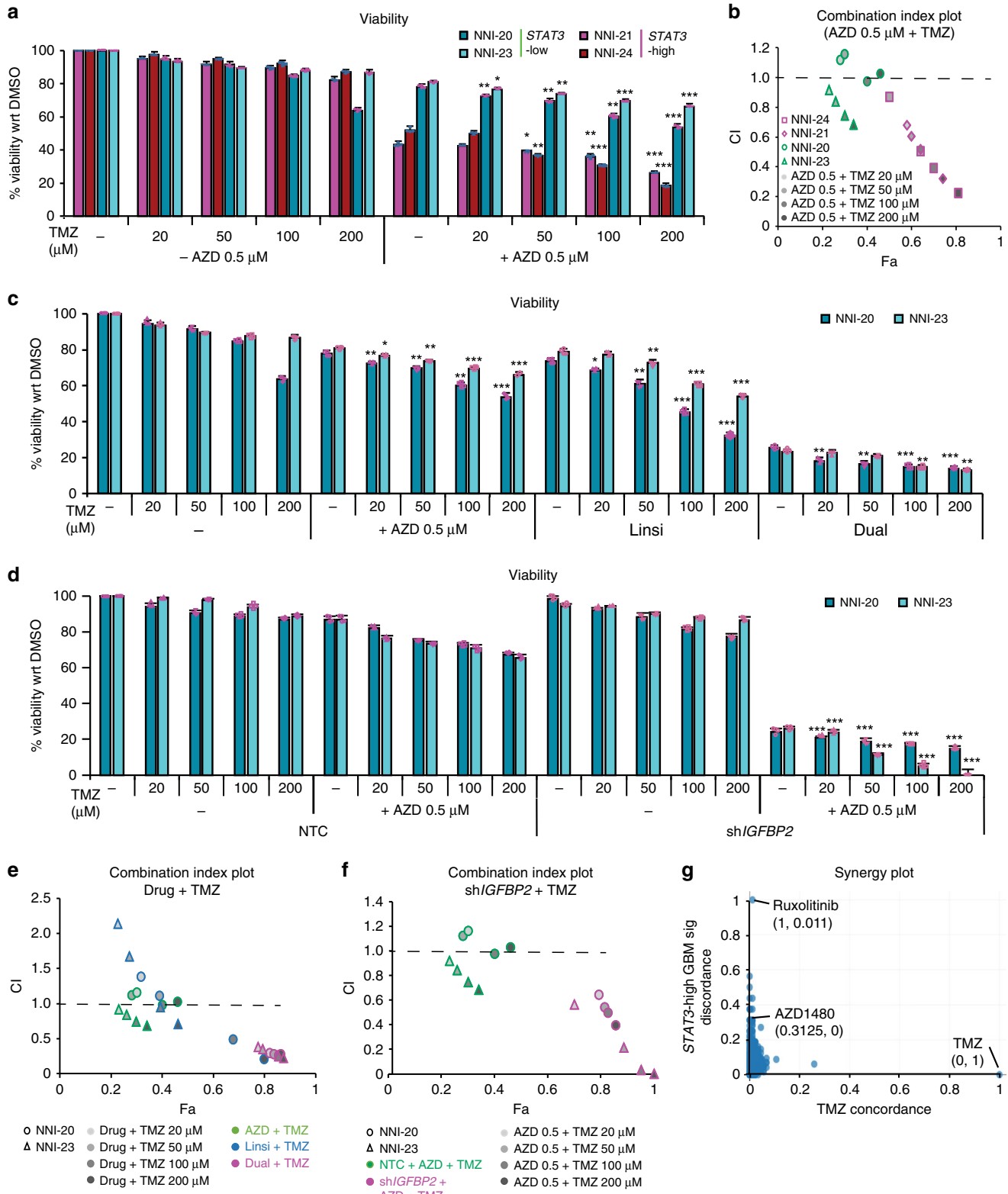

expression of IGFBP2 has been implicated in reduced survival and resistance to chemotherapy[49]. Therefore, our data support that the InsR/IGF-1R pathway may possibly be activated through an autocrine mechanism in a subgroup of GBM tumors. A novel pipeline for analysis of the kinome screen data was implemented in this study, which involved assigning a "biological threshold" to the otherwise voluminous data typical of such screens. The successful application of this method was subsequently confirmed

in vitro and in mouse models implanted with Linsitinib-pretreated cells, a drug targeting IGF-1R. We thus propose a model where STAT3 activation results in binding to nuclear *IGFBP2*, with resultant secretion of IGF-1 cytokine that contributes in a novel feed-forward loop leading to IGF-1R activation. We envisage that this autocrine mechanism can contribute in part to STAT3 activation, since both AZD1480 and Linsitinib dual targeting conferred a significant mitigation of tumor cell

**Fig. 5** Chemosensitization of patient-derived glioblastoma propagating cells (GPCs) with standard-of-care temozolomide treatment. Patient cell lines were treated with increasing doses of temozolomide. **a** The addition of signal transducers and activators of transcription 3 (STAT3) inhibitor AZD1480 to temozolomide treatment demonstrated enhanced chemosensitivity as observed in tumor cell viability. Consistent with bioinformatics prediction, *STAT3*-high cell lines (NNI-21 and NNI-24) displayed enhanced chemosensitivity to AZD1480 treatment with temozolomide, when compared to *STAT3*-low cell lines (NNI-20 and NNI-23). *$p < 0.05$; **$p < 0.01$; ***$p < 0.001$ compared to absence of temozolomide. **b** Combination index (CI)-fraction affected (Fa, indicating fraction of cell viability affected) plots of glioblastoma (GBM) cell lines treated with increasing doses of temozolomide in the presence of 0.5 μM AZD1480. *STAT3*-high cell lines (NNI-21 and NNI-24) displayed a synergistic, cytotoxic effect (CI < 1) with larger Fa, while *STAT3*-low cell lines (NNI-20 and NNI-23) showed marginally reduced Fa values. **c–f** Chemosensitization of *STAT3*-low cell lines (NNI-20 and NNI-23) was observed with temozolomide as demonstrated in the **c**, **d** viability and **e**, **f** CI plot with **e** dual treatment (AZD1480 and Linsitinib) or **f** upon mechanistic gene *IGFBP2* knockdown in combination with 0.5 μM AZD1480. ***$p < 0.001$ sh*IGFBP2* compared to non-targeting control (NTC). In the CI plots, dashed line at CI = 1 indicates an additive effect between two compounds; values above and below indicate antagonism or synergism, respectively. Error bars represent standard deviation of the mean. For statistical analysis, two-sided Student's *t* test was used. **g** Ranking of LINCS compounds (*N* = 1679) based on their concordance with temozolomide consensus signature. Compounds with a high *x* axis value have a signature concordant with temozolomide, and compounds with a high *y* axis value have a signature discordant with the *STAT3*-high GBM disease signature. STAT3 inhibitors, Ruxolitinib and AZD1480, demonstrated low concordance with temozolomide (0.011 and 0, respectively) and high discordance with the *STAT3*-high GBM disease signature (1 and 0.3125, respectively). List of top ranked synergistic compounds able to reverse the *STAT3*-high GBM disease signature is provided in Supplementary Table 4

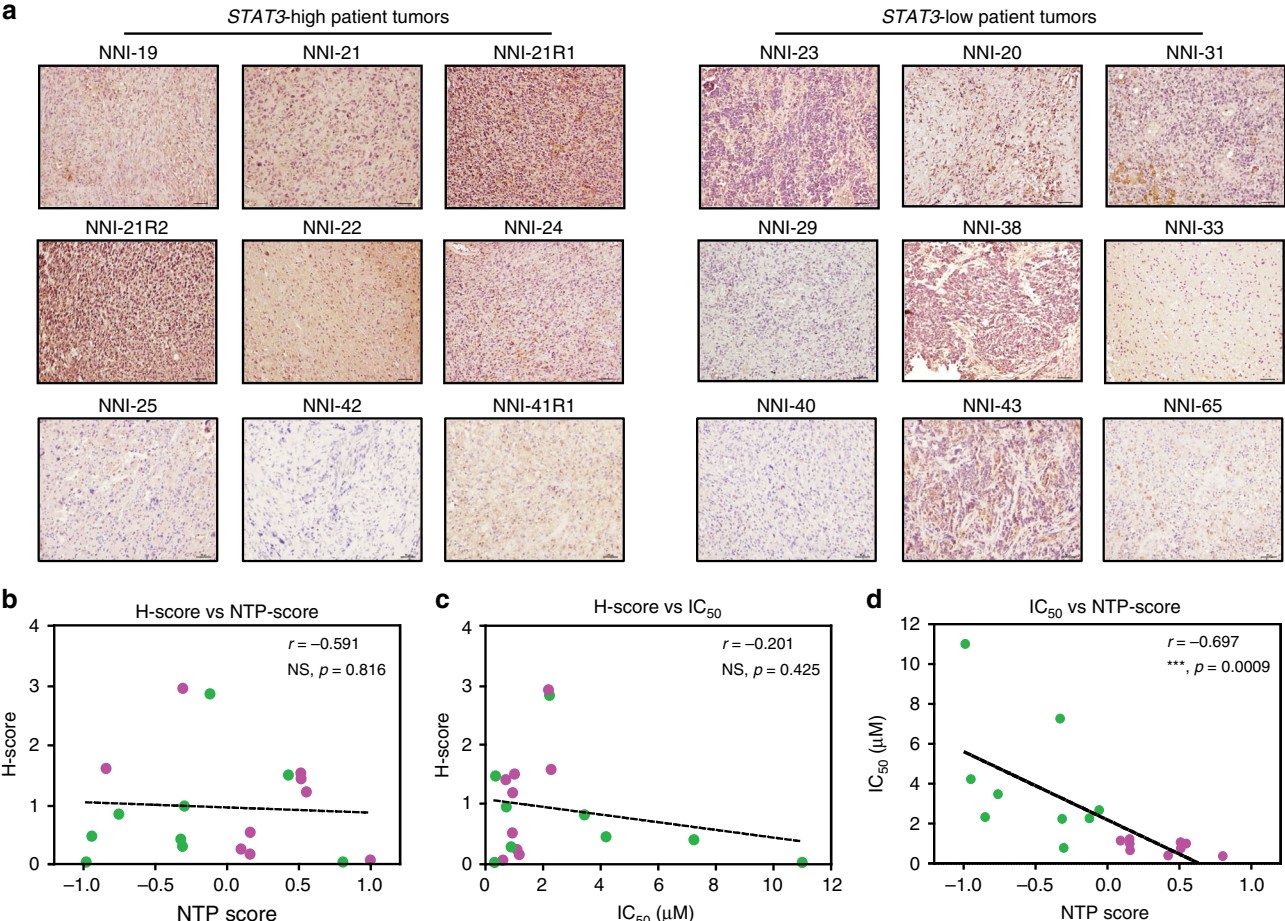

**Fig. 6** NNI-*STAT3* transcriptomic signature better identifies responsive patient cohort. **a** Immunohistochemical (IHC) staining of NNI patient tumors with phospho-signal transducers and activators of transcription 3 (phospho-STAT3). Representative images are shown, scale bar denotes 50 μm. Based on IHC staining, glioblastoma patient tumors (*N* = 18) were not accurately stratified by their *STAT3* status. **b–d** Using three different analyses, there was no significant correlation of **b** *H*-score versus Nearest Template Prediction (NTP) score derived from NNI-*STAT3* signature or **c** *H*-score versus IC$_{50}$, while **d** significant negative correlation was only established when IC$_{50}$ was plotted against the NTP score. ***$p < 0.001$. This indicates that our *STAT3* composite signature accurately identifies the responsive cohort. Magenta dots represent *STAT3*-high values, and green dots represent *STAT3*-low values. For statistical analysis, Pearson correlation coefficient was used

growth and proliferation. As proof-of-concept, we treated *STAT3*-low (NNI-20, -23) and *STAT3*-high (NNI-21, -24) GBM cells with NT157, a selective inhibitor of insulin receptor substrate (IRS-1/2) that has the potential to inhibit both IGF-1R and STAT3 signaling pathways in cancer and stromal cells of the

tumor microenvironment[50]. We observed significant reduction of viability and self-renewal of *STAT3*-low cells, at levels comparable to dual inhibition using AZD1480 and Linsitinib (Supplementary Fig. 9a–c). In contrast, *STAT3*-high cells treated with NT157 demonstrated marginal difference from AZD1480 treatment

alone (Supplementary Fig. 9d, e), suggesting that IGF-1R targeting constitutes no additional benefit in this subgroup. Importantly, we provided evidence that AZD1480 (in STAT3-high GBM cells) and AZD1480/Linsitinib (in STAT3-low GBM cells) synergized with TMZ to mitigate in vitro tumor cell viability. Using transcriptomic information gleaned from clinical and small molecule-treated cell databases, we further identified Ruxolitinib and AZD1480 among the top ranked small molecules capable of reversing the STAT3-high disease transcriptomic profile. While IGF-1R inhibition induces responses as monotherapy in sarcomas and with chemotherapy or targeted agents in common cancers, negative Phase 2/3 trials in unselected patients prompted the cessation of several pharma-led programs[51]. We believe that, with TCGA studies in various cancers, intertumor and intratumor molecular heterogeneity could conceivably play an essential role in patient stratification. Our study suggests the application of IGF-1R and STAT3 inhibition, in combination with TMZ, in STAT3-low GBM tumors.

In both databases using only the GBM cohort, the STAT3 functionally tuned gene signature stratified survival and significantly enriched for the IDH-WT (wild-type) status, suggesting that the latter could act as a clinical molecular indicator for administering STAT3 inhibition therapy (Supplementary Table 1). This would be meaningful as the routine inclusion of the IDH1/2 status is now incorporated into the revised 2016 WHO classification system. We further noted that the IDH-WT cohort consisted of approximately one third of STAT3-low patients in both Gravendeel and TCGA databases (Supplementary Table 1a, contingency table). In this group, wrongful administration of STAT3 inhibitors without prior stratification would lead to the development of resistance, as indicated by our data. Thus we believe that, even though the IDH-WT status is predominantly enriched in the STAT3-high group, the application of the STAT3 composite signature to molecularly subtype the patients remains crucial in the decision to implement STAT3 or STAT3/IGF-1R inhibition therapy in the STAT3-high and -low cohorts, respectively.

We considered the likely scenario of tumor recurrence, typical of the disease's highly infiltrative nature. Our analysis included profiling GBM tumors at first diagnosis and at recurrence, with the finding that mesenchymal STAT3-high tumors largely maintained their molecular profile. In contrast, non-mesenchymal (classical, proneural) tumors underwent molecular switching upon recurrence. In particular, classical tumors at first diagnosis (64%) switched subtypes at recurrence (STAT3-high, 57%; STAT3-low, 43%). This finding has three implications. First, it is imperative that serial molecular profiling be carried out on tumors at all stages to provide a clear decision to the use of STAT3 inhibitory molecules. The failure to stratify patients can potentially result in an unfavorable outcome caused by increased resistance in the STAT3-low cohort. Second, as STAT3 is the key switch effecting PMT, its early implementation when the tumor is STAT3-high and non-mesenchymal could possibly mitigate its subtype switching. Lastly, we suggest that other mechanisms beside EGFR activation can contribute to STAT3 signaling. Recent work by Bonni and colleagues suggested that EGFRvIII-GBM tumors are constitutively active for STAT3, through co-receptor binding of EGFR and OSM[52]. They further postulated that EGFR-wild-type GBM tumors require EGF and OSM cytokines, beside co-receptor binding, to maintain active STAT3 signaling. The classical subtype of GBM tumors is represented by EGFR gain-of-function mutations, such as EGFRvIII[3]. Our earlier observation of subtype switching in classical tumors that were originally STAT3-high thus suggests that additional mechanisms can contribute to STAT3 activation and that combinational therapies may be prescribed. Collectively, our

effort identifies potential drug agents applicable to both STAT3-high and -low patient cohorts.

## Methods

**Materials and cell lines**. GBM tumor specimens from the National Neuroscience Institute (NNI) were obtained with informed consent and de-identified in accordance with the SingHealth Centralised Institutional Review Board A, and GPC culture methods are described below[37–39,53]. All experiments were conducted with low-passage GPCs for which we previously demonstrated maintenance of phenotypic, transcriptomic, and karyotypic features similar to the primary tumor[53]. Briefly, tumors were processed according to Gritti et al. with slight modifications[54]. Cells were seeded at a density of 2500 cells/cm$^2$ in chemically defined serum-free selection growth medium consisting of basic fibroblast growth factor (20 ng/ml; Peprotech Inc., Rocky Hill, NJ), EGF (20 ng/ml; Peprotech Inc.), heparin (5 μg/ml; Sigma-Aldrich, St. Louis, MO), and serum-free supplement (B27; 1×; Gibco, Grand Island, NY) in a 3:1 mix of Dulbecco's modified Eagle's medium (Sigma-Aldrich) and Ham's F-12 Nutrient Mixture (F12; Gibco). The cultures were incubated at 37 °C in a water-saturated atmosphere containing 5% $CO_2$ and 95% air. To maintain the undifferentiated state of neurosphere cultures, growth factors were replenished every 2 days. Successful neurosphere cultures (1–4 weeks) were expanded by mechanical trituration using a flame-drawn glass Pasteur pipette, and cells were reseeded at 100,000 cells/ml in fresh medium.

**Small molecule inhibitors and lentiviral vectors**. Small molecule inhibitor AZD1480 was obtained from SelleckChem and used at concentrations of 0.1, 0.5, 1, or 2 μM. Other STAT3 and IGF-1R small molecule inhibitors were obtained from SelleckChem and used at their respective $IC_{50}$ concentration. TMZ was obtained from Sigma-Aldrich and used at concentrations of 20, 50, 100, and 200 μM. Human lentiviral shRNA clones targeting STAT3 and IGFBP2 in pLKO.1 backbone were from GE Life Science (TRCN0000020840, TRCN0000020842, TRCN0000020843, RHS4080, TRCN0000011033, and TRCN0000006574). Lentiviral shRNA vectors were co-transfected using the Lenti-X HTX Packaging System (Clontech, CA, USA) into HEK293T cells according to the manufacturer's instruction. Viral titer of supernatant collected was determined using Lenti-X™ p24 Rapid Titer Kit (Clontech) according to the manufacturer's instructions. IGF-1R C-terminal-deleted overexpression vectors were constructed using pCDH-CMV-MCS-EF1- GFP+ Puro vector (System Biosciences). The amplified product was digested with XbaI and NotI and ligated into pCDH vector. Lentiviral particles were generated as described above.

**Dose–response curves, viability, and invasion assays**. Dose–response curves and cell viability were assessed using alamarBlue® cell viability assay (Serotec, Oxford, UK) 5 and 10 days post-treatment, respectively[37–39,53]. Dose–response curves for each cell line were generated from a mean of triplicate experiments using GraphPad Prism (GraphPad Software, Inc; USA) and $IC_{50}$ values were computed from 10-point titration curves ranging from $10^{-4}$ to $10^2$ μM. For invasion assay, 50,000 cells were added to the upper compartment of the Corning® BioCoat™ Matrigel® invasion chamber (BD Biosciences, San Jose, CA) and 2% fetal calf serum was supplemented into the lower compartment. Cells were incubated for 24 h and the lower surface was subsequently stained with 0.005% crystal violet (Sigma-Aldrich). The number of cells from five random fields having migrated to the bottom chamber was counted.

**CI values**. CI values based on Loewe's additivity model were determined to assess the nature of drug–drug interactions that can be additive (CI = 1), antagonistic (CI > 1), or synergistic (CI < 1) and effect levels (fraction affected (Fa)). CI and Fa values were calculated using the CompuSyn software (ComboSyn Inc., Paramus, NJ), following the method by Chou et al.[31,32].

**Protein analysis**. Cells were lysed in buffer containing 0.5% sodium deoxycholate, 1% NP-40 detergent, 0.1% sodium dodecyl sulfate (SDS), 0.15 M NaCl, 10 mM Tris-HCl pH7.4, and protease and phosphatase inhibitor cocktail tablets (Roche, Indianapolis, IN). Approximately 25 μg of heat-denatured protein lysate were resolved on 8% SDS polyacrylamide gel and electrotransferred onto polyvinylidene difluoride membranes (Millipore). The following antibodies were used for protein analysis: anti-pSTAT3 (Tyr705; 1:1000; CST, #9138), anti-STAT3 (1:1000; CST, #9139), anti-IGFBP2 (1:1000; CST, #3922), anti-pIGF-1R (Tyr1135/1136; 1:1000; CST, #3024), anti-IGF-1R (1:1000; Santa Cruz, #712), and anti-β-actin (1:10,000; Sigma-Aldrich A5441). Anti-mouse or -rabbit (1:10,000; CST) IgG horseradish peroxidase -linked secondary antibody was used. All antibodies were diluted in 5% bovine serum albumin in 10 mM Tris-HCl pH 7.4, 100 mM NaCl, and 0.1% Tween® 20 (Merck). Membranes were processed per standard procedures and detected using the chemiluminescence detection kit SuperSignal West Pico or Femto (Thermo Scientific) according to the manufacturer's instructions. Protein bands were visualized using SYNGENE G:Box, iChemiXT. Protein expression was quantitated with the Quantity One® software (Bio-Rad Laboratories), normalized against β-actin levels.

**Enzyme-linked immunosorbent assay**. Cells were lysed in buffer containing 0.5% sodium deoxycholate, 1% NP-40 detergent, 0.1% SDS, 0.15 M NaCl, 10 mM Tris-HCl pH7.4, and protease and phosphatase inhibitor cocktail tablets (Roche, Indianapolis, IN). One µg of protein lysates was analyzed on enzyme-linked immunosorbent assay kits as per the manufacturer's protocol in triplicates.

**IDH sequencing**. Genomic DNA was extracted from cell lines and PDXs using the DNeasy Blood & Tissue Kit (Qiagen, Hilden) in accordance with the manufacturer's protocol. PCR amplifications was performed in a total volume of 25 µl with 50 ng of sample (including no template control), using *Pfu* DNA Polymerase (Promega) according to the manufacturer's protocol. Cycle parameters were: initial denaturation at 95 °C for 5 min, followed by 40 cycles of 95 °C for 30 s, 50 °C for 30 s, 72 °C for 90 s, and a final extension at 72 °C for 7 min. PCR products were purified using Wizard® SV Gel and PCR Clean-up System (Promega, USA) and sequenced using the BigDye® Terminator v3.1 Cycle Sequencing Kit (Applied Biosystems, USA). Primers used are as follows:

IDH1 Forward 5′-AATGAGCTCTATATGCCATCACTG-3′;
IDH1 Reverse 5′-TTCATACCTTGCTTAATGGGTGT-3′;
IDH1 sequencing 5′-AATGAGCTCTATATGCCATCACTG-3′;
IDH2 Forward 5′-ATTCTGGTTGAAAGATGGCG-3′;
IDH2 Reverse 5′-CAGAAGAAAGGAAAGCCACG-3′;
IDH2 sequencing 5′-ATTCTGGTTGAAAGATGGCG-3′.

**Immunohistochemistry**. Tissue sections were stained with the following antibodies: anti-pSTAT3 antibody (1:100, CST, #9145) and anti-IGF-1R antibody (1:400, CST, #14534). For quantitative analysis, the percentage of stained tumor cells and intensity of staining were evaluated under high-power fields (×400) on tissue sections using optical microscopy. *H*-scores were then derived from both the staining intensity (scale of 0–3) and the percentage of positive cells (0–100%), generated a score ranging from 0 to 3. Briefly, the percentage of weakly stained cells was multiplied by one plus moderately stained cells multiplied by two plus strongly stained cells multiplied by three. At least five random fields were counted and scoring was performed blinded to clinical data.

**Quantitative real-time reverse transcription PCR**. RNA was isolated using TRI Reagent® (Sigma-Aldrich) and reverse transcribed into cDNA using the Superscript® III First-Strand Synthesis System Kit (Life Technologies). Cycle parameters were: 40 cycles of 95 °C for 10 s, 55 °C for 10 s, and 72 °C for 5 s. Real-time PCR was performed on Roche LightCycler® 96 Instrument using FastStart Essential DNA Green Master (Roche Life Science). Each real-time PCR was done in triplicate, and the level of each gene's expression was determined relative to hypoxanthine phosphoribosyltransferase.

Gene-specific primers are as follows:

STAT3 F-GGGAGAGATTGACCAGCAGT, R-CTGCACTCTCTTCCGGACAT;
ELK3 F-TCAAGACGGAGAAGCTGGAG, R-CCGAGATGAGAAGGGTGAGG;
BIRC2 F-CTCCAGCCTTTCTCCAAACC, R-AGTTACTGAGCTTCCCACCA;
FZD1 F- GCCCTCCTACCTCAACTACC, R-CAGCCGGACAAGAAGATGA;
SLF35F5 F-CTGTGGGGAAACTTACTGCA, R-CCAGTACAACGCCTCCAATG;
KLHDC8A F-CGGGTCTACTGCTCCCTG, R-TGTTGTACATCTCCACGACCT;
GMPPA F-TCACCCAGTTCCTAGAAGCC, R-CTGTTAGCCGTAGTGCCAAG;
SNAP23 F-AGGATGCAGGAATCAAGACCA, R-CTCCACCATCTCCCCATGTT;
NEDD9 F-AGCTCAGGACAAAAGGCTCT, R-GCAACAGCTCCCTTGACAAA;
DTX3L F- TCACAAGCAGAAACACCGTC; R-GTCACCACACACCTTCTCA;
CTNNA1 F-GCAGCCAAAAGACAACAGGA, R-TGTGAGGCATCGTCTGAGG;
NAA38 F-GTCAAGCAGCAAGATGGAGG, R-GCGCATAGTCTTGTTGAGCA;
ITFG3 F-ACACCAACAGCAGCAACAATT, R-AATGAAAGAACTGGGTCTGCC;
IGFBP2 F-GGCTTGGTTGGAAGACTGAT, R-CATTTTCAAAGGCCTCACGC.

**Animal studies**. Mice were handled according to approved guidelines of the Institutional Animal Care and Use Committee of the National Neuroscience Institute, Singapore. Briefly, orthotopic intracranial implantations were carried out using 6–8-week-old NOD-SCID gamma mice (NSG, NOD.Cg-*Prkdcscid Il2rgtm1Wjl*/SzJ, Jackson Laboratories) as described below[37,40]. Five hundred thousand pretreated cells were injected into the following coordinates: antero-posterior = +1 mm; medio-lateral = +2 mm; dorso-ventral = −2.5 mm. Mice were euthanized by transcardiac perfusion with 4% paraformaldehyde upon presentation of neurological deficits. Kaplan–Meier survival curves were plotted to show survival differences. A logrank test was adapted to estimate the survival difference between the STAT3-high and STAT3-low patient group using Prism 5 (GraphPad Software, San Diego, CA). Multivariate Cox Regression model was fitted to identify the significant clinical covariates associated with survival. A *p* value of <0.05 was defined as significant association of covariates for survival. The statistical significance of correlation was evaluated using Spearman's rank correlation test.

**Statistical analysis**. Data are expressed as means ± standard error of the mean (SEM) of at least three independent experiments. Student's *t* or Mann–Whitney *U*

test was used where appropriate. $p \leq 0.05$ was accepted as statistically significant. Survival analyses were performed using the Kaplan–Meier method, with the log-rank test for comparison. $IC_{50}$ values of STAT3 small molecule inhibitors were calculated using nonlinear regression analyses based on dose–response curves. The investigators were not blinded to allocation during experiments and outcome assessment.

**Microarray analysis**. STAT3 knockdown GPCs were profiled on Affymetrix GeneChip® Human Genome U133 Plus 2.0 Array using the 3′ IVT Express Kit. The Gene Expression Omnibus (GEO) accession number for the microarray data is GSE117905. Raw cel files were summarized with mas5 algorithm and $\log_2$-scaled and gene expression dataset was created. All data pre-processing analysis was carried out by R/Bioconductor packages. A linear model was regressed to assess the differentially expressed genes between STAT3 KD and non-targeting control profiles (adjusted *p* value < 0.01) in NNI GPCs ($N = 3$) as described in R/limma packages[27]. False discovery rate (FDR)-adjusted *p* value of <0.05 was considered as statistically significantly perturbed genes upon STAT3 KD. A subset of differential genes was extracted as STAT3 KD gene signature by applying a stringent criterion of $2-\log_2$ fold change between KD clones and the control profiles.

**STAT3 functionally tuned gene signature**. To identify the STAT3 functionally tuned gene signature, we utilized the gene expression data from the Rembrandt glioma patient database ($N = 390$). First, we built a correlation matrix for 44,950 probesets by estimating the pair-wise rank correlation coefficient for each probeset. The STAT3 co-expressed module was defined by the probesets that had a coefficient value >0.3. Both positively and negatively correlated probesets were combined as the STAT3 co-expressed genes in primary tumor samples. The gene list was subsequently narrowed down by selecting only those candidates that showed inverse expression upon STAT3 KD. We considered the intersection of genes upon genetic KD of STAT3 and STAT3 co-regulated transcript modules as the STAT3 functionally tuned gene signature ($N = 207$). The list of genes comprising the STAT3 functionally tuned gene signature is available as Supplementary Data 1.

**Patient stratification**. The evaluation of our STAT3 functionally tuned gene signature was performed on three glioma patient database resources: Gravendeel ($N = 276$), TCGA all glioma patients ($N = 672$), and TCGA GBM microarray cohort ($N = 558$)[3,19,55]. All Affymetrix microarray profiles were processed using standard MAS5 scaling algorithm available in R/affy packages[56]. For TCGA database, gene expression profiles and clinical data were downloaded using R/TCGAbiolinks package[57], and raw cel files for Gravendeel database were downloaded from GEO database (GSE16011). To evaluate the predictive ability of our gene signature, all patient database was treated as an independent validation cohort and was interrogated using the NTP method available in R/CMScaller package[58,59]. The predicted classes for patient tumors with statistical significance (*p* value < 0.05 using 1000 permutation tests) were further evaluated for the prognostic association for overall survival in both glioma or GBM patient cohorts.

**Bioinformatics analysis**. GIS signatures defining the three molecular subgroups were interrogated using single-sample Gene Set Enrichment Analysis (GSEA) with resampling classification strategy[4,60]. To understand the clinical association of 2016 WHO classification marker IDH1 status with STAT3 signature-stratified classes, we employed the relative odds estimation[61]. The relative odds score was estimated from the proportion of highest fifth distribution and lowest fifth distribution of STAT3 signature score values for IDH1 phenotypes. BIC score was calculated for each regression model for survival variability.

**Gene Set Enrichment Analysis**. GSEA was performed using the desktop GSEA software (v.2.2.2) to identify the enrichment of STAT3 KD transcriptome data against established molecular signatures available in public signature databases[62]. We interrogated 186 KEGG pathway gene sets from the Molecular Signatures Database with the complete transcriptome as our background in the enrichment analysis. As our STAT3 KD transcriptome was generated on Affymetrix GeneChip® Human Genome U133 Plus 2.0 Array using 3′ IVT Express Kit, we used max_probe option to collapse the gene expression values of genes with multiple probesets. The overrepresentation of ranked genes score was calculated using weighted running sum statistic option ($p = 2$). The signal-to-noise ratio was applied to rank the genes enriched with the background pathway gene sets using 1000 permutations. The statistical significance of gene ranking between non-target and STAT3 KD clones was defined by the nominal *p* value <0.05. The GSEA ranked list is available as Supplementary Data 2.

**Computational analytical pipeline for kinome data**. We measured the phosphorylation kinetics of 144 kinases in STAT3 signature-stratified GBM cells using the PamChip technology. The computational pipeline is represented in the flowchart (Supplementary Fig. 4b). Briefly, we integrated phosphorylated peptide measurements estimated from PamChip with STAT3 KD transcriptome using published database resources[24–26]. We resolved the multiple peptide-substrate complexity by estimating the pair-wise correlations for every peptide catalogued in

kinase-peptide matrix. The quantitative mean of each peptide cluster was calculated by evaluating the ranks for the presence of large number of correlated peptides having strong expression and dynamic variation across the experimental conditions. We interrogated a linear differential model to estimate the statistical significance between AZD1480- and DMSO-treated cells. A $p$ value of <0.1 was considered as statistically significant. We mapped the AZD1480-altered kinase profiles with *STAT3* KD transcriptomic profiles using two key gene-specific databases[63,64].

**SynergySeq**. We utilized the SynergySeq platform to identify compounds synergistic with TMZ to reverse *STAT3*-high disease signature in GBM patients[33]. The R/shiny package of SynergySeq platform along with drug perturbed signature scores were downloaded from github (https://github.com/schurerlab/SynergySeq; cloned on April 29, 2019). First, we interrogated TCGA GBM patients ($N = 558$) in microarray database with our *STAT3* functionally tuned gene signature using the NTP method available in R/CMScaller package[58,59]. The predicted classes for patient tumors with statistical significance ($p$ value < 0.05 using 1000 permutation tests) were further evaluated to identify the differential disease gene signature. A disease signature of 6359 genes was identified to be differential between *STAT3*-high versus *STAT3*-low GBM patients (FDR $p$ value < 0.0001). This *STAT3*-high GBM signature was interrogated as a disease signature, with TMZ as the reference compound in the SynergySeq pipeline. The LINCS compounds that displayed high disease discordance and low concordance with the reference compound were determined as synergistic small molecule candidates capable of reversing the disease signature based on the Loewe additive model[65]. The current evaluation included compounds from both the LINCS database and GBM-JQ1 study from the SynergySeq project ($N = 1679$).

## Data availability

Data supporting the findings of this work are available within the paper and its Supplementary Information files. A reporting summary for this article is available as a Supplementary Information file. The source data (excel file) containing the list of genes comprising the *STAT3* functionally tuned signature (Fig. 1a), Gene Set Enrichment Analysis (GSEA) ranked gene list (Supplementary Fig. 1m), and winnowed list of genes contributing to chemoresistance (Fig. 3a and Supplementary Fig. 4a, b) are provided. The source data are provided as a Source Data file. The microarray data that support the findings of this study are accessible at the GEO repository, under accession numbers: *STAT3* knockdown microarray data: GSE117905, Gravendeel microarray data: GSE16011, and TCGA molecular data (accession approval required): https://portal.gdc.cancer.gov/projects/TCGA-GBM, https://portal.gdc.cancer.gov/projects/TCGA-LGG.

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

## Acknowledgements

The computational work for this article was partially performed on resources of the National Supercomputing Centre Singapore (https://www.nscc.sg). This research was supported by the Singapore Ministry of Health's National Medical Research Council under its Translational and Clinical Research (TCR) Flagship Programme–Tier 1 (NMRC/TCR/016-NNI/2016) and Clinician Scientist Award (NMRC/CSA-INV/0019/2017) to B.T.A. and an industry award (IRBENVOLEN01) to C.T.

## Author contributions

C.T., B.T.A., and E.S. designed the experiments and developed the methodology. M.S.Y.T., E.S., Y.K.C., S.W.L., L.W.H.K., N.S.T., P.T., B.T.A. and C.T. performed the experiments and collected or analyzed the data. W.H.N. provided clinical material. C.T. and B.T.A. supervised the project, wrote the manuscript, and gave final approval for submission.

## Additional information

**Competing interests:** The authors declare no competing interests.

