## [Peer Review File · Nature Communications]

Reviewers' comments:

Reviewer #1 (Remarks to the Author): Expert in glioma therapy

Despite intensive approaches, including Omics technology, for stratification or molecular biomarkers such as Pten and others etc., glioblastoma multiforme (GBM) is still a devastating disease with poor outcome and almost no survival chance more than two years after diagnosis. In contrast to conventional sequencing approaches the authors in this paper attempt to use a functional approach to identify key pathways with prognostic impact in glioblastoma, which may also provide targets for precision medicine intervention. They provide clear evidence that a STAT3 signaling signature and the IGF receptor pathway do not only bear prognostic impact, but can also be used by inhibitors, e.g. the STAT3 signature, associated with a more aggressive nature of the disease. Using patient samples, tumor spheres from individual patients and a number of functional assays in vitro and in vivo, they provide clear evidence that inhibition of the STAT3 and the IGF receptor pathway, and moreover a combination of both, bear important therapeutic impact. The paper is well written, the experiments are well performed and the focus on functional rather than simply sequencing approaches is certainly very important. The paper has also interesting bioinformatic aspects, making use of large databases.

A few questions remain:

1. In the beginning, the bioinformatic approach should be outlined in more detail. Especially, it is unclear, how the authors come to the inverse expression from STAT3 knockdown in the Rembrandt patient database. Although this could be looked up by the authors, the paper, at least in the supplement, should have more information on that.

2. In Figure 4 restoration of chemosensitivity in STAT3-low glioma cells is shown by IGFBP2 depletion. However in this context, I feel that the term "chemosensitivity" is misleading. In general, "chemosensitivity" means sensitivity towards chemotherapeutic drugs in the sense of classical anticancer drugs. If the authors maintain this term, they should include experiments to address the question of sensitization against the conventional anticancer drugs issued in glioblastoma treatment, i. e. Temozolomide. Also along this line, it might be interesting in general to see, if the inhibition of the STAT3-IGF receptor axis leads to sensitization towards conventional chemotherapy and/or radiation.

3. In Figure 5 the authors show again the issue of sensitization of chemoresistant STAT3-low glioma cells with dual inhibition and state that "NNI-20 (STAT3-low) showed a stark increase in IGF-1R expression". This should be shown in Figure 5 (b). However, in my copy of the manuscript I do not see an increase in IGF-1R expression. This should be clarified.

4. In Figure 6 individual scores of STAT3-high vs. STAT3-low tumors for STAT3 status is shown to which (Fig. 6 b) correlation to the NNI-STAT3 signature and the IC50 status is added. Not surprisingly (Fig. 6 c) high IC50 correlates with a low STAT3 signature score and a high signature score is associated with low IC50. While there is no information on IC50 values in general in the paper, there is also a limited set of patient samples, thus the conclusion, although statistically significant, suffers from low numbers.

5. The approach of targeting the IGF-R1 axis is certainly interesting a valid approach. However, IGF-R targeting has been used in a variety of different tumors, including neuroectodermal tumors such as Ewing's, with mixed results. This should be incorporated at least with one sentence in the discussion.

Reviewer #2 (Remarks to the Author): Expert in Orthotopic models of brain cancer

This is an interesting study evaluating a gene signature for STAT3 that they link to responsiveness

to JAK/STAT inhibitor therapy. The subsequent observation that IGFBP2 is unregulated and mediates resistance to JAK/STAT inhibition in the STAT3-low signature tumors is interesting observation that sets up the combinatorial analysis with a IGF1R inhibitor. The studies are clearly presented and appear carefully performed. The results are interesting and potentially provide a rationale to pursue such combinatorial strategies, although the extent of benefit observed in difference in median survival is only approximately 20% (see Figure 5C). The other issues with the manuscript are listed below.

Given the strong correlation between low STAT3 signature and IDH mutant and low grade tumors, the analysis in Figure 1 should be repeated in a more homogenous population of IDH1/2 wild-type GBM. The IDH mutation status should be reported for all of the PDX models being analyzed.

The term 'multiforme' is no longer part of the GBM diagnostic name and should be dropped from the manuscript.

The dosing regimen in Figure 2 for the mice is not described, and the nomenclature of reporting drug concentration instead of dose in the legend is not informative.

Throughout the legends, the number of independent experiments used to generate the data displayed should be noted. This is especially relevant for the western blotting results, which are not described in the methods section.

Figure 5A/B: generally, evaluation of synergy requires a formal analysis using isobologram analysis, combination index evaluation, or other analytic strategy and is evaluated with across multiple concentrations of either drug used in alone or in combination.

Figure 5C/D: as above, the dosing regimen is unclear with similar issues regarding the concentration vs. dose of drug. The claim that AZD treatment results in a stark increase in IGF-1R levels in NNI-20 tumors is unfounded. There is a small variation. While not noted, if each lane is from a single animal, these differences could readily be explained through typical variation across individual tumors. To make a solid claim regarding upregulation of IGF1R, biological replicate samples for each treatment condition would be typical.

Reviewer #3 (Remarks to the Author): Expert in bioinformatics

Si Yan Tan et al. develop a signature associated to the activation of STAT3 in GBM patients and show its diagnostic properties. They also show a combined STAT3 signature analysis with kinome screen data on STAT3 inhibitor-treated cells.

The topic of the paper is important and the role of STAT3 in the mesenchymal transformation of brain tumors have been considered in several studies such as Carro et al. (Nature 2010) where the authors definitely show that STAT3 is the main initiator and master regulator of mesenchymal transformation of IDH-WT GBM and is associated with poor survival.

One of the main concerns on this paper is the way in which the authors derive the signature and how they state its prognostic properties in clinical dataset such as Rembrandt, TCGA and Gravendeel. I will focus on the computational part, as it is the pre-requisite for all the results they present in the experimental part.

I believe that the paper has some major weaknesses in this section.

First of all, the authors claim that their signature "stratifies GBM patients into STAT3-high and -low cohorts". GBM are defined as grade IV glioma, however the datasets they use for the evaluation of the signature include grade II, III and IV patients.

-Figure 1. the derivation of the signature as explained in Figure 1 and in the methods needs to be better defined. My understanding is that the authors intersect the differentially expressed genes after KD of STAT3 with the genes co-expressed with STAT3 in Rembrandt. A detailed explanation of how they identify co-expressed genes is needed. However, as it was quite obvious, the STAT3 signature divides grade IV tumors and lower grade tumors, and hence the survival difference in Figure 1c. Note that the "N" in the figures showing differential expression should be number of sample not the number of different genes.

- Figure 1. The authors should select just grade IV and show the prognostic abilities of the signature. Table 2 and Supplementary Figure 2 associated to the main Figure 1 are very confusing. Supplementary Figure 2b shows perfect association between STAT3-high and IDH-wild-type glioma, whereas STAT3-low tumors are IDH-mutant in the TCGA. I deduce that they are using both the LGG cohort and the GBM cohort of the TCGA. The GBM is microarray, the LGG is RNAseq, could the authors explain how they did the merging of the two datasets? Moreover Supplementary Figure 2b reports $175+175=350$ samples. Which cohort of 350 samples are they using? If they use the LGG RNAseq cohort, it should be more than 570 samples, whereas the GBM microarray cohort is also more than 400 samples. The multivariate analysis in Supplementary table 2, where the authors want to state the prognostic value of the signature, is not convincing if the authors do not clearly specify which cohort they are using, how many samples, are they all GBM? Or do they include LGG and GBM? The multivariate analysis also shows that IDH mutation alone (in the TCGA) is a better predictor of survival than their signature. This is not the case in the Gravendeel dataset. How do the authors explain this different behavior.

Why the authors do not use Rembrandt in supplemental table 2? Clinical information for the Rembrandt dataset is also available. There are approaches to predict IDH status from gene expression, if they want to do multivariate analysis.

- Supplementary figure 1. Panel D reports GSEA enrichment in "in STAT3 knockdown expression profile revealed JAK-STAT pathway depletion in (a) Rembrandt and (b) Gravendeel clinical glioma databases". This is very confusing. What is the ranked list and what is the gene-set of this analysis? This is not the way to present an enrichment analysis reporting some biological insight of a comparison between different conditions.

Reviewer 1 (expert in glioma therapy)

Despite intensive approaches, including Omics technology, for stratification or molecular biomarkers such as Pten and others etc., glioblastoma multiforme (GBM) is still a devastating disease with poor outcome and almost no survival chance more than two years after diagnosis. In contrast to conventional sequencing approaches the authors in this paper attempt to use a functional approach to identify key pathways with prognostic impact in glioblastoma, which may also provide targets for precision medicine intervention. They provide clear evidence that a STAT3 signaling signature and the IGF receptor pathway do not only bear prognostic impact, but can also be used by inhibitors, e.g. the STAT3 signature, associated with a more aggressive nature of the disease. Using patient samples, tumor spheres from individual patients and a number of functional assays in vitro and in vivo, they provide clear evidence that inhibition of the STAT3 and the IGF receptor pathway, and moreover a combination of both, bear important therapeutic impact. The paper is well written, the experiments are well performed and the focus on functional rather than simply sequencing approaches is certainly very important. The paper has also interesting bioinformatic aspects, making use of large databases.

A few questions remain:

- 1. In the beginning, the bioinformatic approach should be outlined in more detail. Especially, it is unclear, how the authors come to the inverse expression from STAT3 knockdown in the Rembrandt patient database. Although this could be looked up by the authors, the paper, at least in the supplement, should have more information on that.*

Response

We agree with the need for clearer description. We now incorporate the derivation of the *STAT3* functionally-tuned gene signature in the Methods section, on **Page 33** of the tracked manuscript. Briefly, to identify the *STAT3* functionally-tuned gene signature, we utilized the gene expression data from the Rembrandt glioma patient database (N=390 patients). First, we built a correlation matrix for 44,950 probesets by estimating the pair-wise rank correlation coefficient for each probeset. The *STAT3* co-expressed module was defined by the probesets that had a coefficient value greater than 0.3. Both positively and negatively correlated probesets were combined as the *STAT3* co-expressed genes in primary tumor samples. The gene list was subsequently narrowed down by selecting only those candidates that showed inverse expression upon *STAT3* knockdown. We considered the intersection of genes upon genetic knockdown of *STAT3* and *STAT3* co-regulated transcript modules as the functionally-tuned *STAT3* signature (N=207 genes).

We rationalize that using Rembrandt, the clinically relevant genes may be playing passenger roles even though they are co-expressed with *STAT3*. Thus, to identify candidates that are downstream of *STAT3* and yet must demonstrate phenotypic effects, we intersected the gene list from Rembrandt with those generated from *STAT3* knockdown (which must be inverted upon a phenotypic outcome). The *STAT3* knockdown phenotype was separately verified using viability and self-renewal assays (**Supplementary Fig. 1**).

- 2. In Figure 4 restoration of chemosensitivity in STAT3-low glioma cells is shown by IGF2BP2 depletion. However in this context, I feel that the term “chemosensitivity” is misleading. In general, “chemosensitivity” means sensitivity towards chemotherapeutic drugs in the sense of classical anticancer drugs. If the authors maintain this term, they should include experiments to address the question of sensitization against the conventional anticancer drugs issued in glioblastoma treatment, i. e. Temozolomide. Also along this line, it might be interesting in general to see, if the inhibition of the STAT3-IGF receptor axis leads to sensitization towards conventional chemotherapy and/or radiation.*

Response

We acknowledge that the reviewer has raised an important issue on chemosensitization towards the standard of care drug, temozolomide (TMZ). We have reorganized **Fig. 4** to reflect the efficacy of the dual inhibition strategy (STAT3 and IGF-1R) in *STAT3*-low cells; whilst removing the word “chemosensitivity”. We have also made related amendments in the text to reflect this consistency in **Fig. 4**.

We noted similar feedback from Reviewer 2, comment #6, on demonstrating synergism using combination index plots or isobologram analysis with regards to “chemosensitivity”. We now include new data of how our proposed STAT3 inhibition therapy, as well as STAT3 and IGF-1R dual inhibition compare with TMZ. The data are now captured in a new **Fig. 5**. Briefly, we treated *STAT3*-high and –low cells with AZD1480 with/without TMZ at 20-200 μ M, an *in vitro* concentration range common in literature [**Fig. 5A(a)**]¹⁻³. We observed that *STAT3*-high cells demonstrated significant reduction in viability that was synergistic with TMZ at 50-200 μ M. We evaluated the combination index (CI) plot where increased synergism with TMZ correlated with CI values of less-than-1 [**Fig. 5A(b)**]. The CI values were calculated using the CompuSyn software for evaluation of drug combinations^{4,5}.

Next, in response to the reviewer’s interest in the STAT3-IGF-1R dual inhibition strategy, we assessed the fraction affected (i.e. reduced viability) in *STAT3*-low cells after treatment with AZD1480, Linsitinib or both [**Fig. 5B(a)**]. Similarly, our CI plot showed increasing synergism with TMZ, as denoted by the decreasing CI value [**Fig. 5C(a)**].

We also carried out similar AZD1480 and TMZ treatment in *STAT3*-low cells with *IGFBP2* knockdown for definitive mechanistic implication [**Fig. 5B(b)**]. The rationale arose from our earlier data showing that STAT3 activation leads to *IGFBP2* gene transcription, which in turn stimulates the production of IGF cytokine (**Supplementary Fig. 6**). Our CI plot demonstrated synergism with TMZ at 50-200 μ M range [**Fig. 5C(b)**].

Collectively, our *in vitro* data provides strong evidence for both STAT3 inhibition and dual STAT3/IGF-1R inhibition in *STAT3*-high and –low cells respectively; and synergize with TMZ, thus suggesting the advancement of both therapeutic approaches in a clinical setting.

To provide further support of our *in vitro* data above, we focused on demonstrating that STAT3 inhibitors can selectively target *STAT3*-high glioblastoma (GBM) tumors. The premise of our approach lies in The Cancer Genome Atlas (TCGA) studies showing that gene expression drives GBM disease progression and prognostic outcome. We then tapped into a recent article where drug and disease signature integration identifies synergistic combinations in GBM⁶. This study utilized the Library of Integrated Network-Based Cellular Signatures (LINCS) database where several commercial cancer cell lines were treated with FDA-approved and experimental small molecule drugs, and the transcriptomic profile of each treated cell line was acquired. Primary GBM cells were similarly treated with these drugs including TMZ with radiation. The authors further mapped the association of transcriptomic patterns to prognostic information in TCGA, thus identifying clinically relevant drug combinations capable of reversing the disease transcriptomic profile. In our specific scenario, the disease pattern is defined by our *STAT3* functionally-tuned gene signature. Thus, using an orthogonal plot, we identified drugs that demonstrated low concordance with TMZ, and high discordance with the *STAT3*-high tumor phenotype [**Fig. 5C(c)**]. This indicates that the drug is synergistic with TMZ, and is capable of reversing the disease gene signature profile specified by the *STAT3* composite signature. Interestingly, Ruxolitinib, a Jak2 inhibitor, and AZD1480 emerged in the top ranked drugs (**Supplementary Table 7**). Ruxolitinib is currently in clinical trial for GBM. These drugs thus have the potential to reverse the *STAT3*-high disease profile, and supports their use in targeting the proneural-mesenchymal transition (PMT) that typifies highly aggressive and recurrent tumors.

3. *In Figure 5 the authors show again the issue of sensitization of chemoresistant STAT3-low glioma cells with dual inhibition and state that “NNI-20 (STAT3-low) showed a stark increase in IGF-1R expression”. This should be shown in Figure 5 (b). However, in my copy of the manuscript I do not see an increase in IGF-1R expression. This should be clarified.*

Response

We have consolidated data surrounding the sensitization of STAT3-low cells using a dual inhibition therapeutic approach in an updated **Fig. 4**. Briefly, **Fig. 4D(b)** now shows an increase in IGF-1R protein expression in NNI-20 cells when treated with AZD1480. This expression was abolished when the cells were treated with an IGF-1R inhibitor, Linsitinib. We clarify that the immunoblot quantification represents 2 mouse brains derived from the animal experiments in **Fig. 4C(b)**. This is due to only 2 brains from the dual treatment arm that were sizeable enough for retrieval. We emphasize that **Fig. 4C(b)** demonstrated significant survival benefit conferred by the dual inhibition treatment, thus possibly accounting for the difficulty in getting material for immunoblot analysis.

4. *In Figure 6 individual scores of STAT3-high vs. STAT3-low tumors for STAT3 status is shown to which (Fig. 6 b) correlation to the NNI-STAT3 signature and the IC50 status is added. Not surprisingly (Fig. 6 c) high IC50 correlates with a low STAT3 signature score and a high signature score is associated with low IC50. While there is no information on IC50 values in general in the paper, there is also a limited set of patient samples, thus the conclusion, although statistically significant, suffers from low numbers.*

Response

We agree with the reviewer and have now expanded to an additional 3 primary tumors for each of STAT3-high and STAT3-low subtypes (**updated Fig. 6**). All related glioma cell IC₅₀ value derivation is now shown in a new **Supplementary Fig. 7**. We recognize that although we are still statistically underpowered with a total of 18 primary tumors, we emphasize that within each tumor, there was significant correlation between the IC₅₀ value of each successfully-established primary cell line, and gene signature score of the original primary tumor.

5. *The approach of targeting the IGF-R1 axis is certainly interesting a valid approach. However, IGF-R targeting has been used in a variety of different tumors, including neuroectodermal tumors such as Ewing's, with mixed results. This should be incorporated at least with one sentence in the discussion.*

Response

We have now incorporated some discussion surrounding the use of IGF-1R inhibitors in other malignancies, including neuroectodermal tumors, on **Page 23** of the tracked manuscript. Briefly, while IGF-1R inhibition induces responses as monotherapy in sarcomas and with chemotherapy or targeted agents in common cancers, negative Phase 2/3 trials in unselected patients prompted the cessation of several pharma-led programs⁷. We believe that with TCGA studies in various cancers, inter- and intra-tumor molecular heterogeneity could conceivably play an essential role in patient stratification – the principles of precision oncology. Our study suggests the application of IGF-1R and STAT3 inhibition, in combination with TMZ, in STAT3-low GBM tumors.

We thank Reviewer 1 for the insightful feedback and substantial improvement of the manuscript.

Reviewer 2 (expert in orthotopic models of brain cancer)

1. *This is an interesting study evaluating a gene signature for STAT3 that they link to responsiveness to JAK/STAT inhibitor therapy. The subsequent observation that IGF1R is unregulated and mediates resistance to JAK/STAT inhibition in the STAT3-low signature tumors is interesting observation that sets up the combinatorial analysis with a IGF1R inhibitor. The studies are clearly presented and appear carefully performed. The results are interesting and potentially provide a rationale to pursue such combinatorial strategies, although the extent of benefit observed in difference in median survival is only approximately 20% (see Figure 5C). The other issues with the manuscript are listed below.*

Response

The previous Fig. 5C has now been re-organized into an updated **Fig. 4C**. We noted that in both animal models established from *STAT3*-high [**Fig. 4C(a)**] and *STAT3*-low cells [**Fig. 4C(b)**] pretreated with AZD1480, Linsitinib or both drugs, the median survival difference is more than 20%, comparing AZD1480 with DMSO in *STAT3*-high tumors; or comparing dual AZD1480/Linsitinib with single agent or DMSO alone. We hope to increase enthusiasm with our newly acquired data on “chemosensitivity” in a new **Fig. 5**, where in both *STAT3*-linked tumor subtypes, the drugs synergize with TMZ to further mitigate glioma cell viability.

2. *Given the strong correlation between low STAT3 signature and IDH mutant and low grade tumors, the analysis in Figure 1 should be repeated in a more homogenous population of IDH1/2 wild-type GBM. The IDH mutation status should be reported for all of the PDX models being analyzed.*

Response

We agree with the reviewer on this important comment and have now re-organized **Fig. 1**. Briefly, **Figs. 1F, G** now show new analyses in only GBM tumors in the Gravendeel database. Analyses in TCGA database with only GBM tumors is shown in an updated **Supplementary Fig. 2**. Contingency tables, univariate and multivariate analyses are shown in **Supplementary Table 3**. In both databases using only the GBM cohort, the *STAT3* functionally-tuned gene signature stratified survival and significantly enriched for the *IDH*-WT (wild-type) status, suggesting that the *IDH*-WT status could act as a clinical molecular indicator for administering *STAT3* inhibition therapy. This would be meaningful as the routine inclusion of the *IDH1/2* status is now incorporated into the revised WHO classification system.

We noted that the *IDH*-WT cohort consisted of approximately one-third of *STAT3*-low patients in both Gravendeel and TCGA databases (**Supplementary Table 3(a)**, contingency table). In this group, wrongful administration of *STAT3* inhibitors without prior stratification would lead to the development of resistance, as indicated by our data. Thus, we believe that even though the *IDH*-WT status is predominantly enriched in the *STAT3*-high group, the application of the *STAT3* gene signature to molecularly subtype the patients remains crucial in the decision to implement *STAT3*, or *STAT3*/*IGF-1R* inhibition therapy in the *STAT3*-high and -low cohorts, respectively. We have now explained this in the Discussion section, **Page 23-24**.

The *IDH* mutation status for all our PDX models is now included in an **updated Supplementary Fig. 3E**, where NNI-20, 21, 23, 24 cells and NNI-20, 24 PDX tumors are wild-type for *IDH1/2*.

3. *The term 'multiforme' is no longer part of the GBM diagnostic name and should be dropped from the manuscript.*

Response

We agree with the reviewer and have dropped all “multiforme” words from the revised manuscript. We apologize for our oversight.

4. *The dosing regimen in Figure 2 for the mice is not described, and the nomenclature of reporting drug concentration instead of dose in the legend is not informative.*

Response

We wish to clarify that all mouse models in the manuscript were established from patient-derived glioma cells pretreated with small molecule drugs or vehicle control (originally reflected as “pretreated” on **Page 15** of tracked manuscript). The concentrations used in the drug treatment were stated in all figures in the original submission. For the avoidance of doubt, we have improved our wording in the Methods section, **Page 31** of tracked manuscript to include the word “pretreated”. We have also specifically amended text in the main article to reflect that all animal models were created using pretreated GBM cells. We apologize for the confusion.

5. *Throughout the legends, the number of independent experiments used to generate the data displayed should be noted. This is especially relevant for the western blotting results, which are not described in the methods section.*

Response

We apologize for our oversight. We have now included all such details in the updated figure legends.

6. *Figure 5A/B: generally, evaluation of synergy requires a formal analysis using isobologram analysis, combination index evaluation, or other analytic strategy and is evaluated with across multiple concentrations of either drug used in alone or in combination.*

Response

We agree with the reviewer and have re-organized the original data into the new **Fig. 4B** and **Supplementary Fig. 5B**. We calculated using CompuSyn, CI = 0.2092 for the drug combination, 1 μ M AZD1480 and 0.5 μ M Linsitinib in NNI-20, and CI = 0.23 in NNI-23.

We highlight that Reviewers 1 and 2 shared the same important comment on drug synergy and alluded to “chemosensitization”. For ease of reading, we will copy below our reply to Reviewer 1, comment #2:

We noted similar feedback from Reviewer 2, comment #6, on demonstrating synergism using combination index plots or isobologram analysis, and we now elaborate more with regards to “chemosensitivity”. We now include new data of how our proposed STAT3 inhibition therapy, as well as STAT3 and IGF-1R dual inhibition compare with TMZ. The data are now captured in a new **Fig. 5**. Briefly, we treated *STAT3*-high and -low cells with AZD1480 with/without TMZ at 20-200 μ M, concentration range common in literature [**Fig. 5A(a)**]¹⁻³. We observed that *STAT3*-high cells demonstrated significant reduction in viability that was synergistic with TMZ at 50-200 μ M. We evaluated the combination index (CI) plot where increased synergism with TMZ correlated with CI

values of less-than-1 [Fig. 5A(b)]. The CI values were calculated using the CompuSyn software for evaluation of drug combinations^{4,5}.

Next, in response to the reviewer's interest in the STAT3-IGF-1R dual inhibition strategy, we assessed the fraction affected (i.e. reduced viability) in *STAT3*-low cells after treatment with AZD1480, Linsitinib or both [Fig. 5B(a)]. Similarly, our CI plot showed increasing synergism with TMZ (increasingly negative ratio) [Fig. 5C(a)].

We also carried out similar AZD1480 and TMZ treatment in *STAT3*-low cells with *IGFBP2* knockdown for definitive mechanistic implication [Fig. 5B(b)]. The rationale arose from our earlier data showing that STAT3 activation leads to *IGFBP2* gene transcription, which in turn stimulates the production of IGF cytokine (Supplementary Fig. 6). Our CI plot demonstrated synergism with TMZ at 50-200 μ M range [Fig. 5C(b)].

Collectively, our *in vitro* data provides strong evidence for both STAT3 inhibition and dual STAT3/IGF-1R inhibition in *STAT3*-high and -low cells respectively; and synergize with TMZ, thus suggesting the advancement of both therapeutic approaches in a clinical setting.

To provide further support of our *in vitro* data above, we focused on demonstrating that STAT3 inhibitors can selectively target *STAT3*-high glioblastoma (GBM) tumors. The premise of our approach lies in The Cancer Genome Atlas (TCGA) studies showing that gene expression drives GBM disease progression and prognostic outcome. We then tapped into a recent article where drug and disease signature integration identifies synergistic combinations in GBM⁶. This study utilized the Library of Integrated Network-Based Cellular Signatures (LINCS) database where several commercial cancer cell lines were treated with FDA-approved and experimental small molecule drugs, and the transcriptomic profile of each treated cell line was acquired. Primary GBM cells were similarly treated with these drugs including TMZ with radiation. The authors further mapped the association of transcriptomic patterns to prognostic information in TCGA, thus identifying clinically relevant drug combinations capable of reversing the disease transcriptomic profile. In our specific scenario, the disease pattern is defined by our *STAT3* functionally-tuned gene signature. Thus, using an orthogonal plot, we identified drugs that demonstrated low concordance with TMZ, and high discordance with the *STAT3*-high tumor phenotype [Fig. 5C(c)]. Interestingly, Ruxolitinib, a Jak2 inhibitor, and AZD1480 emerged in the top 10 ranked drugs (Supplementary Table 7). Ruxolitinib is currently in clinical trial for GBM. These drugs thus have the potential to reverse the *STAT3*-high disease profile, and supports their use in targeting the proneural-mesenchymal transition (PMT) that typifies highly aggressive and recurrent tumors.

7. *Figure 5C/D: as above, the dosing regimen is unclear with similar issues regarding the concentration vs. dose of drug. The claim that AZD treatment results in a stark increase in IGF-1R levels in NNI-20 tumors is unfounded. There is a small variation. While not noted, if each lane is from a single animal, these differences could readily be explained through typical variation across individual tumors. To make a solid claim regarding upregulation of IGF1R, biological replicate samples for each treatment condition would be typical.*

Response

We clarify that all mouse models were established from orthotopic xenografts using pretreated GBM cells (see Comment #4 above), originally reflected as "pretreated" on Page 15 of tracked manuscript. The concentrations used in the drug treatment were stated in all figures in the original submission. For the avoidance of doubt, we have improved our wording in the Methods section, Page 31 of tracked manuscript to include the word "pretreated". We have also specifically amended text in the main article to reflect that all animal models were created using pretreated GBM cells. We apologize for the confusion.

We noted a similar comment from Reviewer 1, comment #3, on IGF-1R protein levels in NNI-20. We copy our previous response below for ease of reading:

We have consolidated data surrounding the sensitization of *STAT3*-low cells using a dual inhibition therapeutic approach in an updated **Fig. 4**. Briefly, **Fig. 4D(b)** now shows an increase in IGF-1R protein expression in NNI-20 cells when treated with AZD1480. This expression was abolished when the cells were treated with an IGF-1R inhibitor, Linsitinib. We clarify that the immunoblot quantification represents 2 mouse brains derived from the animal experiments in **Fig. 4C**. This is due to only 2 brains from the dual treatment arm that were sizeable enough for retrieval. We emphasize that **Fig. 4C(b)**, Set 1, demonstrated significant survival benefit conferred by the dual inhibition treatment, thus possibly accounting for the difficulty in getting material for immunoblot analysis (Set 2, **Supplementary Fig. 5E**).

We thank Reviewer 2 for the insightful feedback and substantial improvement of the manuscript.

Reviewer 3 (expert in bioinformatics)

Si Yan Tan et al. develop a signature associated to the activation of STAT3 in GBM patients and show its diagnostic properties. They also show a combined STAT3 signature analysis with kinome screen data on STAT3 inhibitor-treated cells.

The topic of the paper is important and the role of STAT3 in the mesenchymal transformation of brain tumors have been considered in several studies such as Carro et al. (Nature 2010) where the authors definitely show that STAT3 is the main initiator and master regulator of mesenchymal transformation of IDH-WT GBM and is associated with poor survival.

One of the main concerns on this paper is the way in which the authors derive the signature and how they state its prognostic properties in clinical dataset such as Rembrandt, TCGA and Gravendeel. I will focus on the computational part, as it is the pre-requisite for all the results they present in the experimental part. I believe that the paper has some major weaknesses in this section.

- 1. First of all, the authors claim that their signature "stratifies GBM patients into STAT3-high and -low cohorts". GBM are defined as grade IV glioma, however the datasets they use for the evaluation of the signature include grade II, III and IV patients.*

Response

We thank Reviewer 3 for this important feedback. We have now carried out the analyses in only GBM patient cohorts of Gravendeel and TCGA databases. The *STAT3* functionally-tuned gene signature stratified the overall survival of GBM patients in Gravendeel (log-rank p -value=0.002) and TCGA (log-rank p -value=0.009) databases. We show the additional survival plots for GBM patients in the revised manuscript (**Gravendeel, Figs. 1F, G; TCGA, Supplementary Figs. 2C, D**). In addition, the univariate and multivariate analyses for GBM patients are included in the revised version [**Supplementary Table 3(b)**]. To make the signature-based stratification consistent across patient tumors and glioma cell databases, we adapted the Nearest Template Prediction (NTP) algorithm using 1,000 permutations throughout our analyses. Detailed methodologies for the stratification and patient numbers are described in our tracked manuscript, **Page 34**.

- 2. Figure 1. the derivation of the signature as explained in Figure 1 and in the methods needs to be better defined. My understanding is that the authors intersect the differentially expressed genes after KD of STAT3 with the genes co-expressed with STAT3 in Rembrandt. A detailed explanation of how they identify co-expressed genes is needed. However, as it was quite obvious, the STAT3 signature divides grade IV tumors and lower grade tumors, and hence the survival difference in Figure 1c. Note that the "N" in the figures showing differential expression should be number of sample not the number of different genes.*

Response

We now clarify our methodology details in the updated Methods section, **Page 33** of tracked manuscript. We first extracted *STAT3* co-expressed gene modules from Rembrandt glioma patients (N=390) by identifying the genes that had the correlation coefficient $\pm 0.3 <$ with the *STAT3* probeset. The co-expressed gene module was further intersected with the *STAT3* knockdown signature. The *STAT3* functionally-tuned signature includes those positively-correlated genes that were decreased and, those negatively-correlated genes that were significantly increased upon *STAT3* knockdown perturbation. This would allow us to identify clinically relevant genes acting downstream of *STAT3*.

We recognize the importance of demonstrating the signature performance in exclusively GBM patients, to avoid confounding data due to tumor grade. The data are as described above in

Comment #1. The prognostic association of the *STAT3* functionally-tuned gene signature in GBM patients has been amended in the tracked Results section, **Page 9** [*“Similar prognostic association was observed in GBM-only patient databases for *STAT3*-high and -low cohorts (Gravendeel, log-rank p -value=0.002; TCGA, log-rank p -value=0.009)”*]. We now amend “N” to represent sample numbers in **Fig. 1**.

3. *Figure 1. The authors should select just grade IV and show the prognostic abilities of the signature. Table 2 and Supplementary Figure 2 associated to the main Figure 1 are very confusing. Supplementary Figure 2b shows perfect association between *STAT3*-high and IDH-wild-type glioma, whereas *STAT3*-low tumors are IDH-mutant in the TCGA. I deduce that they are using both the LGG cohort and the GBM cohort of the TCGA. The GBM is microarray, the LGG is RNAseq, could the authors explain how they did the merging of the two datasets? Moreover Supplementary Figure 2b reports 175+175=350 samples. Which cohort of 350 samples are they using? If they use the LGG RNAseq cohort, it should more than 570 samples, whereas the GBM microarray cohort is also more than 400 samples. The multivariate analysis in Supplementary table 2, where the authors want to state the prognostic value of the signature, is not convincing if the authors do not clearly specify which cohort they are using, how many samples, are they all GBM? Or do they include LGG and GBM? The multivariate analysis also shows that IDH mutation alone (in the TCGA) is a better predictor of survival than their signature. This is not the case in the Gravendeel dataset. How do the authors explain this different behavior?*

Responses splitted up according to questions within above paragraph

“Figure 1. The authors should select just grade IV and show the prognostic abilities of the signature.”

Please see response to Comments #1 and #2 above.

*“Table 2 and Supplementary Figure 2 associated to the main Figure 1 are very confusing. Supplementary Figure 2b shows perfect association between *STAT3*-high and IDH-wild-type glioma, whereas *STAT3*-low tumors are IDH-mutant in the TCGA.”*

We wish to clarify that the data presented in Fig. 1 is from Gravendeel (N=276), whilst Supplementary Fig. 2 is additional data from TCGA RNA-Seq data (N=392) from the University of North Carolina database. Table 2 describes the univariate and multivariate results for both databases. We agree with the reviewer’s observation that the enrichment of *IDH*-status with *STAT3* signature classes was variable between the two patient databases. To address this, we collected the complete RNA-Seq data (N=672) from the recent package release from R/TCGAbiolinks⁸. We evaluated the proportion of WHO classification markers between TCGA RNA-Seq and Gravendeel patient databases and found that there was a significant difference in distribution (shown in figure below). This could account for the differential enrichment (Fisher’s exact p -value=0.0113). In addition, we performed Bayesian Information Criterion (BIC) analysis on several covariates to identify the model contributing to prognostic variability, using a step-wise regression model-building analysis [**Fig. 1D(a)**]. The results demonstrate that the compound model including *STAT3* stratification classes, on top of WHO classification markers, significantly improves the base model to account for the survival variability in the patient database [Δ BIC=44.72 points; p -value<0.01; please see 3rd bar from bottom and 4th bar from the top in **Fig. 1D(a)**].

Figure Legend: Bar plot distribution of WHO classification markers in Gravendeel (N=276) and TCGA RNA-Seq glioma patient (N=672) databases. *IDH*, isocitrate dehydrogenase; mut, mutant; wt, wild-type; CDL, 1p/19q co-deleted; NCDL, 1p/19q non-co-deleted.

“I deduce that they are using both the LGG cohort and the GBM cohort of the TCGA. The GBM is microarray, the LGG is RNAseq, could the authors explain how they did the merging of the two datasets? Moreover, Supplementary Figure 2b reports 175+175=350 samples. Which cohort of 350 samples are they using? If they use the LGG rnaseq cohort, it should more that 570 samples, whereas the GBM microarray cohort is also more than 400 samples.”

We wish to clarify that we used all glioma patient databases which included both LGG and GBM samples from Gravendeel microarray (N=276 tumors; GEO accession number: GSE16011) and TCGA RNA-Seq database (N=392 including 241 LGG patients and 151 GBM patients). TCGA RNA-Seq database was extracted from the University of North Carolina RNA-Seqv2 level 3 data resource from TCGA bundle manifest file on 09/20/2013. The *STAT3* signature interrogation was performed in patient databases using the connectivity map (c-map) algorithm in our original submission. This c-map approach failed to assign *STAT3* signature classes for 42 patients (Kolmogorov-Smirnov test p -value>0.05, non-significant), which achieved the total of 350 patients with *STAT3*-high, -low status. As suggested by the reviewer, we collected TCGA RNA-Seq data for 672 patients using R/TCGAbiolinks package and the results from the updated TCGA RNA-Seq database are now incorporated in the revised version [**Supplementary Figs. 2C, D; Supplementary Tables 3(a) and 3(b)**]. In addition, our GBM patient analysis was executed separately on N=558 microarray normalized data, downloaded using the R/TCGAbiolinks package. Our methods about patient stratification include the details of patient numbers and the computational approaches used in the revised version, **Page 34**. We also confirm that we did not merge the TCGA RNA-Seq database with the GBM microarray database in our analyses.

“The multivariate analysis in Supplementary table 2, where the authors want to state the prognostic value of the signature, is not convincing if the authors do not clearly specify which cohort they are using, how many samples, are they all GBM? Or do they include LGG and GBM? The multivariate analysis also shows that IDH mutation alone (in the TCGA) is a better predictor of survival than their signature. This is not the case in the Gravendeel dataset. How do the authors explain this different behavior?”

We clarify that our original analyses were conducted in all glioma patient databases. **Supplementary Tables 3(a) and 3(b)** in the revised version specifically mention “All Glioma” and “GBM” patients in the headers. The differential behavior with regards to the *IDH* mutation status between the Gravendeel and TCGA databases may be due to the variable distribution of *IDH* status as described above (see Figure).

- 4. Why the authors do not use Rembrandt in supplemental table 2? Clinical information for the Rembrandt dataset is also available. There are approaches to predict IDH status from gene expression, if they want to do multivariate analysis.*

Response

We clarify that our *STAT3* functionally-tuned gene signature was derived from Rembrandt primary tumors, and NNI glioma cell *STAT3* knockdown transcriptomic information. To validate the predictive ability of our signature, we evaluated in two independent clinical databases including Gravendeel and TCGA. In addition, we assessed both the *IDH* mutation and chromosomal 1p/19q co-deletion status as described in the revised 2016 WHO classification scheme in our multivariate analysis. As recommended by the reviewer, we predicted the *IDH* status using an immunological gene signature⁹ in Rembrandt and observed similar results, as in other databases, see below **Table**.

Interestingly, the *STAT3* signature status was the sole predictor for overall survival in the GBM cohort (HR=1.538; log-rank *p*-value=0.025).

Covariates	All Glioma					
	Univariate analysis			Multivariate analysis		
	HR (95% CI)	SE	Pr(> z)*	HR (95% CI)	SE	Pr(> z)*
STAT3 -High	3.084 (2.296-4.142)	0.15	7.5e-14	2.139 (1.497-3.057)	0.18	3.0e-5
W.H.O IDH -WT	2.476 (1.746-3.511)	0.18	3.7e-7	1.382 (0.876-2.178)	0.23	0.164
GI Classical	2.767 (1.945-3.935)	0.18	1.5e-8	1.744 (1.119-2.719)	0.23	0.014
GI Mesenchymal	3.201 (2.276-4.503)	0.17	2.3e-11	1.662 (1.048-2.636)	0.24	0.031
Age	1.018 (1.008-1.027)	0.01	2.0e-4	1.02 (1.01-1.03)	0	6.4e-5

Covariates	GBM					
	Univariate analysis			Multivariate analysis		
	HR (95% CI)	SE	Pr(> z)*	HR (95% CI)	SE	Pr(> z)*
STAT3 -High	1.538 (1.056-2.24)	0.19	0.025	1.538 (1.056-2.24)	0.19	0.025
W.H.O IDH -WT	1.187 (0.699-2.014)	0.27	0.527	N.S.		
GI Classical	1.179 (0.711-1.955)	0.26	0.524			
GI Mesenchymal	1.243 (0.784-1.971)	0.24	0.356			

Table: Univariate and multivariate analysis of *STAT3* signature classes, glioma-intrinsic (GI) molecular subtypes, *IDH* mutation status (predicted) and age. *STAT3*-low patient cohort, *IDH*-mutation, GI-PN were considered as reference categories to estimate the coefficients in a Cox regression model. HR = hazard ratio; Pr(<|z|) = two-sided Wald test *p*-value; SE = standard error of coefficient; WT = wild-type; GI = glioma-intrinsic. N.S., not significant.

5. *Supplementary figure 1. Panel D reports GSEA enrichment in "in STAT3 knockdown expression profile revealed JAK-STAT pathway depletion in (a) Rembrandt and (b) Gravendeel clinical glioma databases". This is very confusing. What is the ranked list and what is the gene-set of this analysis? This is not the way to present an enrichment analysis reporting some biological insight of a comparison between different conditions.*

Response

We now amend the GSEA plot, showing that the non-targeting transcriptome profile was positively enriched with the JAK/STAT pathway module from the KEGG molecular database, compared with STAT3 knockdown glioma cell clones. A collection of 186 KEGG molecular pathway gene sets from the Molecular Signatures Database (MSigDB) was evaluated against the complete transcriptome as background. The ranked gene list of the JAK/STAT pathway from KEGG gene sets which includes 155 genes is presented as **Supplementary Table 2** in the revised manuscript.

We thank Reviewer 3 for the insightful feedback and substantial improvement of the manuscript.

Overall Comments

We sincerely thank all reviewers for their insightful feedback, and hope to be considered for publication in Nature Communications. Our team has been, honestly, humbled by the review process. We constantly remind ourselves to rise to the standards and challenges posed by high-ranking, reputable journals. We hope that we have presented a significant effort in this revision.

References

1. Garnier, D. *et al.* Divergent evolution of temozolomide resistance in glioblastoma stem cells is reflected in extracellular vesicles and coupled with radiosensitization. ***Neuro-oncology*** 20, 236-248 (2018).
2. Liu, N., Hu, G., Wang, H., Li, Z. & Guo, Z. PLK1 inhibitor facilitates the suppressing effect of temozolomide on human brain glioma stem cells. ***Journal of cellular and molecular medicine*** 22, 5300-5310 (2018).
3. Ulasov, I. V., Nandi, S., Dey, M., Sonabend, A. M. & Lesniak, M. S. Inhibition of Sonic Hedgehog and Notch Pathways Enhances Sensitivity of CD133+ Glioma Stem Cells to Temozolomide Therapy. ***Molecular Medicine*** 17, 103-112 (2011).
4. Chou, T.-C. Theoretical Basis, Experimental Design, and Computerized Simulation of Synergism and Antagonism in Drug Combination Studies. ***Pharmacological Reviews*** 58, 621 (2006).
5. Chou, T.-C. Drug Combination Studies and Their Synergy Quantification Using the Chou-Talalay Method. ***Cancer Research*** 70, 440 (2010).
6. Stathias, V. *et al.* Drug and disease signature integration identifies synergistic combinations in glioblastoma. ***Nat Commun*** 9, 5315 (2018).
7. King, H., Aleksic, T., Haluska, P. & Macaulay, V. M. Can we unlock the potential of IGF-1R inhibition in cancer therapy? ***Cancer Treat Rev*** 40, 1096-1105 (2014).
8. Colaprico, A. *et al.* TCGAAbiolinks: an R/Bioconductor package for integrative analysis of TCGA data. ***Nucleic Acids Res*** 44, e71 (2016).
9. Mu, L. *et al.* The IDH1 Mutation-Induced Oncometabolite, 2-Hydroxyglutarate, May Affect DNA Methylation and Expression of PD-L1 in Gliomas. ***Front Mol Neurosci*** 11, 82 (2018).

REVIEWERS' COMMENTS:

Reviewer #1 (Remarks to the Author):

The authors have addressed all critical points raised in the previous review appropriately. Thus, the manuscript has further improved.

Reviewer #2 (Remarks to the Author):

The authors have adequately addressed this reviewer's concerns.

Reviewer #3 (Remarks to the Author):

The revised manuscript clarified my original concerns.